# New particle formation in the volcanic eruption plume of the Piton de la Fournaise: specific features from a long-term dataset

Clémence Rose[1], Brice Foucart[2], David Picard[1], Aurélie Colomb[1], Jean-Marc Metzger[3], Pierre Tulet[2], Karine Sellegri[1]

[1]Laboratoire de Météorologie Physique (LaMP-UMR 6016, CNRS, Université Clermont Auvergne), 63178, Aubière, France.
[2]Laboratoire de l'Atmosphère et des Cyclones (LACy-UMR 8015, CNRS, Université de La Réunion, Météo France), 97744 Saint Denis de La Réunion, France.
[3]Observatoire des Sciences de l'Univers de La Réunion, UMS 3365 (CNRS, Université de La Réunion), 97744, Saint Denis de La Réunion, France.

*Correspondence to*: Clémence Rose (c.rose@opgc.univ-bpclermont.fr) and Karine Sellegri (k.sellegri@opgc.univ-bpclermont.fr)

## Abstract

New particle formation (NPF) is a key atmospheric process which may be responsible for a major fraction of the total aerosol number burden at the global scale, including in particular cloud condensation nuclei (CCN). NPF has been observed in various environments around the world, but some specific conditions, such as those encountered in volcanic plumes, remain poorly documented in the literature. Yet, understanding such natural processes is essential to better define preindustrial conditions and their variability in climate model simulations. Here we report observations of NPF performed at the high-altitude observatory of Maïdo (2165 m a.s.l., La Réunion Island) between 1$^{st}$ January and 31$^{st}$ December 2015. During this time period, 3 effusive eruptions of the Piton de la Fournaise, located ~ 39 km away from the station, were observed and documented, resulting in 29 days of measurement in volcanic plume conditions to be compared with 250 "non-plume days". This dataset is, to our knowledge, the largest ever reported for the investigation of NPF in tropospheric volcanic plume conditions, and allowed for the first time a statistical approach to characterize the process and also assess its relevance with respect to non-plume conditions. NPF was observed on 90% of the plume days vs 71% of the non-plume days during the 4 months when the eruptions occurred. The events were on average detected earlier on plume days, most likely benefiting from larger amounts of precursors available at the site prior to nucleation hours. The overall effect of the plume conditions on the particle growth rate was limited. However, with the exception of September, particle formation rates were significantly higher on plume days. The signature of the volcanic plume on the aerosol spectra up to $d_p = 600$ nm was further investigated based on the analysis and fitting of the particle size distributions recorded in and off-plume conditions. The spectra recorded prior to nucleation hours, in absence of freshly formed particles, featured a significant contribution of particles likely formed *via* heterogeneous processes at the vent of the volcano (and assimilated to volcanic primary particles) to the concentrations of the 2 accumulation modes on plume days. Later on in the morning, the concentrations of the nucleation and Aitken modes showed important variations on plume days compared to event days outside of plume conditions. The spectra recorded on event days, in and off-plume conditions, were further used to provide an average size distribution of the particles of volcanic origin, which clearly highlighted the

dominant contribution of secondary over primary particles (93%) to the total concentration measured on NPF event days within volcanic plume. In a next step, particular attention was paid to the concentration of particles with $d_p > 50$ nm ($N_{50}$), used as a proxy for potential CCN population. The contribution of secondary particles to the increase of $N_{50}$ was the most frequent in plume conditions, and the magnitude of the increase was also more important on plume days compared to non-plume days.

Last, in order to further evaluate the effect of volcanic plume conditions on the occurrence of NPF, we analysed the variations of the condensation sink (CS) and [$H_2SO_4$], previously reported to play a key role in the process. Over the investigated months, higher CS (calculated prior to nucleation hours) were observed in plume conditions, and coincided with high $SO_2$ mixing ratios. Those most likely compensated for the strengthened loss rate of the vapours and favoured the occurrence of NPF, suggesting at the same time a key role of $H_2SO_4$ in the process. This last hypothesis was further supported by the correlation

between the formation rate of 2 nm particles ($J_2$) and [$H_2SO_4$], and by the fair approximation of $J_2$ that was obtained by the mean of a recent parameterisation of the binary nucleation of $H_2SO_4 - H_2O$. This last result demonstrates that in absence of direct measurement of [$H_2SO_4$] and sub-3 nm particles concentration, estimates of $J_2$ could be fairly estimated from the knowledge of $SO_2$ mixing ratios only. Finally, the use of the parameterisation for ion induced binary nucleation also highlighted the likely significant contribution of ion induced nucleation for [$H_2SO_4$] below $\sim 8 \times 10^8$ cm$^{-3}$.

**1 Introduction**

Aerosol particles are a complex component of the atmospheric system which affects both air quality and climate. They have been the focus of a growing number of studies during the last decades, but our knowledge of their sources, properties, including their ability to interact with other atmospheric components and associated effects on the Earth's climate system, remains nonetheless uncomplete. In specific, while particles are known to affect the formation of clouds, and in turn their properties

(Albrecht, 1989; Rosenfeld et al., 2014), the radiative forcing associated to these effects (usually referred to as "indirect effect") is known with a still large uncertainty (Myhre et al., 2013). Better understanding and quantification of this indirect effect requires, in particular, more accurate information on secondary aerosol particle sources, and in turn on new particle formation (NPF). Indeed, measurements conducted in various environments suggest that NPF might be an important source of cloud condensation nuclei (CCN) (e.g. Kerminen et al., 2012; Rose et al., 2017), which is further supported by model investigations

(Merikanto et al., 2009; Makkonen et al., 2012; Gordon et al., 2017). However, despite significant improvement of instrumental techniques for the characterisation of the newly formed particles and their precursors (Junninen et al., 2010; Vanhanen et al., 2011; Jokinen et al., 2012), model predictions are still affected by our limited understanding of NPF. In addition, the scarcity of observations makes it all the more uncertain in hard-to-reach environments, or in specific conditions, such as those encountered in volcanic plumes.

Volcanic eruptions are one of the most important natural sources of some specific gases and aerosol particles in the atmosphere. A variety of gaseous species have been identified in volcanic plumes, among which halogens (Aiuppa et al., 2009; Mather, 2015) and sulphur dioxide ($SO_2$), which can be further oxidized to sulfuric acid ($H_2SO_4$). $SO_2$ is released in significant amount,

in particular during eruptive periods but also from passive degassing (Andres and Kasgnoc, 1998; Robock, 2000; Tulet et al., 2017), and it was reported that all together, volcanoes contribute significantly to the global sulphur budget, which is otherwise dominated by anthropogenic sources (Penner et al., 2001 and references therein; Seinfield and Pandis, 2006). Volcanic aerosols are injected into the atmosphere both as primary or secondary particles. The former can be fragments of ash which, despite

their relatively large sizes (up to few microns) (Robock, 2000), can be transported over long distances in the atmosphere (Hervo et al., 2012). Primary sulphate aerosols of volcanic origin were also evidenced by near source measurements conducted at Masaya volcano by Allen et al. (2002), who were, however, not able to identify their precise mechanisms of formation. Several pathways were later suggested for the formation of $H_2SO_4$ at the vent, including catalytic oxidation of $SO_2$ inside the volcanic dome (Zelenski et al., 2015), high temperature chemistry in the gas phase (Roberts et al., 2019), as well as aqueous production

of $H_2SO_4$ from $SO_2$ (Tulet et al., 2017). $H_2SO_4$ produced by the mean of the aforementioned reactions is expected to contribute to the formation and growth of particles in the close vicinity of the volcano, which are in turn assimilated to primary volcanic aerosols. Secondary particles, on the other hand, result from gas-to-particle conversion processes, including NPF, that take place outside of the volcanic dome under atmospheric temperature and pressure.

The occurrence of NPF in volcanic plume conditions was suspected to take place in several earlier studies (e.g. Deshler et al.,

1992; Robock, 2000; Mauldin et al., 2003), but the first dedicated study was conducted by Boulon et al. (2011), during the eruption of the Eyjafjallajokull which happened in spring 2010. Indeed, using measurements performed at the high-altitude station of puy de Dôme (1465 m a.s.l., France), the authors linked the occurrence of NPF to unusually high levels of $H_2SO_4$ corresponding to model predictions of the Eyjafjallajokull plume transport to puy de Dôme, and highlighted a remarkably elevated particle formation rate in these conditions. While $H_2SO_4$ concentrations were derived from a proxy in this last study,

Sahyoun et al. (2019) recently reported new evidence and quantification of the NPF process in the passive plumes of Etna and Stromboli, and supported the key role of $H_2SO_4$ using direct measurement performed with a state-of-the-art mass spectrometer onboard the French research aircraft ATR-42. The rarity of studies dedicated to the observation of NPF in volcanic plume conditions is illustrated by the absence of any related topic in the recent review by Kerminen et al. (2018), despite the need for understanding such natural processes. Indeed, those might have dominated NPF and CCN formation in the pristine

preindustrial era, when anthropogenic emissions were much lower. Our incomplete knowledge of the preindustrial conditions is responsible for a significant fraction of the uncertainty on the impact of aerosols on climate, since these conditions form the baseline to calculate the radiative forcing caused by anthropogenic emissions in climate model simulations (Carslaw et al, 2013; Gordon et al, 2016, 2017). In specific, substantial uncertainties in the pre-industrial baseline cloud radiative state related to the activity of continuously degassing volcanoes were reported by Schmidt et al. (2012).

Despite providing new and highly valuable information, the studies by Boulon et al. (2011) and Sahyoun et al. (2019), however, had some limitations. Indeed, they were both based on short datasets, which did not allow for any statistical approach to evaluate the relevance of the process nor proper comparison with the occurrence of NPF outside of plume conditions. Also, airborne measurements conducted in the close vicinity of Etna and Stromboli allowed Sahyoun et al. (2019) to investigate the presence of the newly formed particles soon after the emission of their precursors from the vent of the volcanoes up to few

tens of kilometres. They were, however, unfortunately not able to document properly the evolution of the particle size distribution along the volcanic plumes, but analysed instead the particle concentration in relatively broad size ranges (2.5 – 10 nm, 10 – 250 nm). In addition, as mentioned earlier, this study was focussed on passive plumes, i.e. in the presence of a limited concurrent emission of primary particles by the volcanoes. Conducting similar investigation of volcanic eruption plumes is by

the way more difficult due to the unexpected aspect of active eruptions. Taking the advantage of ground-based measurements and broader instrumental setup, Boulon et al. (2011) were in contrast able to study the time variation of the particle size distribution between 2 nm and 20 µm, and, in turn, to evaluate the strength of the reported NPF events in terms of particle formation and growth rates. Measurements were, however, conducted after the volcanic eruption plume of Eyjafjallajokull had travelled several thousands of kilometres, and most likely underwent significant modifications due to the occurrence of

chemical processes and dilution during transport.

In this context, the objectives of the present work were to provide new observations of NPF in a volcanic eruption plume with detailed analysis of the event characteristics, including the capacity of the newly formed particles to reach CCN sizes, and to assess the relevance of the process with respect to non-plume conditions. For that purpose, we used measurements of the particle number size distribution and $SO_2$ mixing ratios conducted at the Maïdo observatory (2165 m a.s.l., La Réunion Island,

Baray et al., 2013) between January $1^{st}$ and December $31^{st}$ 2015. During this period, 3 eruptions of the Piton de la Fournaise, which is located in the south-eastern sector of the island (see Fig. 1.a from Tulet et al., 2017), were observed and documented, resulting in 29 days of measurement in plume conditions that could be compared with 250 "non-plume days". This dataset is, to our knowledge, the largest ever reported for the investigation of NPF in volcanic plume conditions.

## 2. Measurements and methods

### 2.1 Measurements

Measurements were performed at the Maïdo observatory located on La Réunion Island, in the Indian Ocean (21.080° S, 55.383° E) between January $1^{st}$ and December $31^{st}$ 2015. This high-altitude station (2165 m a.s.l.) is surrounded by lush tropical vegetation on the east side (natural amphitheatre of Mafate) and highland tamarin forest on the west side, which dominates the coast where the nearest urban areas of Saint Paul and Le Port are located (105 000 and 40 000 inhabitants, 13 and 15 km away

from the station, respectively). The observatory was built in 2012, and since then it has been progressively equipped with a growing instrumental setup dedicated to the monitoring of the upper troposphere and the lower stratosphere, including both in-situ and remote sensing techniques. Measurements conducted at Maïdo are of great interest since the observatory is one of the very few multi-instrumented stations in the tropics, and more particularly in the Southern hemisphere. Evidence of this is the involvement of the station in several international networks such as NDACC (Network for the Detection of Atmospheric

Composition Change) and ACTRIS (Aerosol Cloud and Trace gases Research Infrastructure), and in the GAW (Global Atmospheric Watch) regional network, which all ensure the quality of the data collected at this site. In addition, the proximity of the Piton de la Fournaise, located in the south-eastern region of the island ~ 39 km away from Maïdo, gives a strategic

position to the station for the particular investigation of volcanic plume conditions, which are in the scope of the present work. More detailed information about the facility can be found in Baray et al. (2013) and Foucart et al. (2018), including a description of the large and local scale atmospheric dynamics which affect the observations performed at the site. An overview of the monitoring of the volcanic plume performed at Maïdo during the eruptions of the Piton de la Fournaise observed in 2015 is in addition available in Tulet et al. (2017).

The instrumental setup used in the present work was previously described in the companion study by Foucart et al. (2018). The aerosol size distribution between 10 and 600 nm was measured with a custom-built Differential Mobility Particle Sizer (DMPS), with a time resolution of 5 min. Particles are first charged to equilibrium using a Ni-63 bipolar charger, after which they enter the DMPS, which includes a TSI-type Differential Mobility Analyzer (DMA) operating in a closed loop and a Condensation Particle Counter model TSI 3010. The instrument was operated behind a Whole Air Inlet (higher size cut-off of 25 µm for an average wind speed of 4 m s$^{-1}$), and measurement protocols were defined with respect to the ACTRIS recommendations regarding both the flow rates and RH (Wiedensohler et al., 2012). The particle size distributions measured with the DMPS were used to detect the occurrence of NPF and to calculate the event characteristics, such as the particle formation and growth rates (see Sect. 2.2), and to further evaluate the potential of the newly formed particles to reach CCN relevant sizes. DMPS data was also used to calculate the condensation sink (CS), which describes the loss rate of gaseous precursors on pre-existing particles (Kulmala et al., 2012). In addition to DMPS, an Air Ion Spectrometer (Airel, Estonia, Mirme et al., 2007) was operated at Maïdo between May and October 2015. The AIS includes two DMA which allow simultaneous detection of negative and positive ions and charged particles in the mobility range 0.0013-3.2 cm$^2$ V$^{-1}$ s$^{-1}$, corresponding to mobility diameter 0.8-42 nm in NTP-conditions (Mäkelä et al., 1996). Each analyser is working with a flow rate of 90 L min$^{-1}$ (sample flow rate of 30 L min$^{-1}$ and sheath flow rate of 60 L min$^{-1}$) in order to reduce diffusion losses in the instrument, and measurements were conducted through an individual short inlet (30 cm) to further limit the loss of ions in the sampling line. AIS data was collected with a time resolution of 5 min and was analysed in the present work to get further insight into the timing of the early stages of the NPF process.

SO$_2$ mixing ratios used to monitor the occurrence of volcanic plume conditions at Maïdo were measured with a UV fluorescence analyser (Thermo Environmental Instrument TEI 43) operated by ATMO-Réunion (formerly referred to as ORA, Observatoire Réunionnais de l'Air), with a time resolution of 15 min. The detection limit of the instrument was about 0.5 ppb, which is above the usual SO$_2$ mixing ratios encountered at Maïdo outside of the eruptive periods of the Piton de la Fournaise (see Fig. A1 in Foucart et al. 2018).

Finally, meteorological parameters recorded with a time resolution of 3 s were used as ancillary data. Global radiation was measured with a Sunshine Pyranometer (SPN1, Delta-T Dvices Ltd., resolution 0.6 W m$^{-2}$), and other variables of interest including temperature, wind speed and direction and relative humidity (RH) were recorded using a Vaisala Weather Transmitter WXT510.

An overview of the data availability for all abovementioned instruments between January 1$^{st}$ and December 31$^{st}$ 2015 is provided in Foucart et al. (2018, Fig. 2).

## 2.2 Particle formation and growth rates

As previously reported in Foucart et al. (2018), the formation rate of 12 nm particles ($J_{12}$) expressed in cm$^{-3}$ s$^{-1}$ was calculated based on DMPS data following Kulmala et al. (2012):

$$J_{12} = \frac{dN_{12-19}}{dt} + CoagS_{12} \times N_{12-19} + \frac{GR_{12-19}}{7\,nm} \times N_{12-19} \tag{1}$$

where $N_{12-19}$ (cm$^{-3}$) is the number concentration of particles in the diameter range 12-19 nm, $CoagS_{12}$ (s$^{-1}$) represents the coagulation sink of 12 nm particles on larger particles and $GR_{12-19}$ (nm h$^{-1}$) is the particle growth rate between 12 and 19 nm. The second and third terms on the right-hand side of Eq. (1) thus represent the loss rate of particles in the range 12-19 nm due to coagulation on pre-existing larger particles and condensational growth outside of the size range, respectively. Together with the production term $J_{12}$, these loss terms determine the time evolution of the particle concentration in the range 12-19 nm, denoted as $\frac{dN_{12-19}}{dt}$. $GR_{12-19}$ was estimated from DMPS data based on the "Maximum method" introduced by Hirsikko et al. (2005). Briefly, the times $t_m$ when the maximum concentration successively reached each of the DMPS size bins between 12 and 19 nm were first determined by fitting a normal distribution to the concentration time series of each bin. $GR_{12-19}$ was then obtained by fitting a linear least square fit through the $t_m$ values previously identified. Note that the use of $CoagS_{12}$ for all particles in the range between 12 and 19 nm might cause some overestimation of the actual coagulation losses, and in turn lead to high estimates of $J_{12}$.

In order to get further insight into the early stages of the NPF process, and in absence of direct measurement of sub-3 nm particles, $J_{12}$ and $GR_{12-19}$ were used to derive the formation rate of 2 nm particles ($J_2$) following the equation from Lehtinen et al. (2007):

$$J_2 = \frac{J_{12}}{exp\left(-\gamma \times 2\,nm \times \frac{CoagS_2}{GR12-19}\right)} \tag{2}$$

where,

$$\gamma = \frac{1}{m+1}\left[\left(\frac{12\,nm}{2\,nm}\right)^{m+1} - 1\right] \tag{3}$$

and,

$$m = \frac{log(CoagS_{12}) - log(CoagS_2)}{log(12\,nm) - log(2\,nm)} \tag{4}$$

In Eq. (2) and (4), $CoagS_2$ represents the coagulation sink of 2 nm particles on larger particles. Equation (2) is based on the assumption that the particle growth rate is constant over the range 2 – 19 nm, which is most likely not the case, as particle growth rates are usually reported to increase with particle diameter (Yli-Juuti et al., 2011), including high altitude sites (e.g. Rose et al., 2015). In order to further investigate the effect of the particle growth rate on the prediction of $J_2$ with Eq. (2), we have performed a sensitivity study. The term $\gamma \times 2\,nm \times CoagS_2$ (hereafter referred to as Fact$_J$) was found to vary between

~ $5.0 \times 10^{-5}$ and ~ $1.2 \times 10^{-3}$ nm s$^{-1}$ (10[th] and 90[th] percentiles, respectively), and the $GR_{12-19}$ obtained during the events of interest, i.e. observed in May, August, September and October (see Sect 2.3 for more details about the selected period), were found in the range between $GR_{low}$ = 7.8 nm h$^{-1}$ and $GR_{high}$ = 42.7 nm h$^{-1}$ (10[th] and 90[th] percentiles, respectively). Considering the abovementioned lower limit of Fact$_J$, varying $GR_{12-19}$ between $GR_{low}$ and $GR_{high}$ did not affect the calculation of $J_2$, and repeating the same procedure with $GR_{low}/2$ and $GR_{high} \times 2$ lead to the same result. Considering then a case with Fact$_J$ in the high end of observed values, i.e. ~ $1.2 \times 10^{-3}$ nm s$^{-1}$, varying $GR_{12-19}$ between $GR_{low}$ and $GR_{high}$ resulted in a factor of ~ 1.6 in the calculation of $J_2$, and repeating the test with the more extreme values $GR_{low}/2$ and $GR_{high} \times 2$ only increased the factor to ~ 3.0. All in all, this sensitivity study demonstrates that the accuracy of the particle growth rate only has a limited effect on the calculation of $J_2$ with Eq. (2) in the conditions of the present analysis.

## 2.3 Detection of the volcanic plume

Four eruptions of the Piton de la Fournaise were observed in 2015: the first in February, the second in May, the third at the very beginning of August, and, the last, from the end of August to late October. More details about the exact dates and characteristics of the eruptions can be found in Tulet et al. (2017). Figure 1.a presents the timeseries of the SO$_2$ mixing ratio measured at Maïdo between May and December 2015, and highlights as well the eruptive periods of the Piton de la Fournaise observed during this period (Tulet et al., 2017), which logically coincide with the detection of high SO$_2$ levels at the station. More specifically, the Maïdo was considered to be in volcanic plume conditions when at least three of the hourly averages of the SO$_2$ mixing ratio measured between 06:00 and 11:00 LT (UTC +4) were ≥1 ppb, which corresponds to the 97th percentile of SO$_2$ mixing ratio on non-eruptive days. The relatively low SO$_2$ mixing ratios observed outside of the eruptive periods, mostly below the detection limit of the instrument, reflect the low pollution levels characteristic of this insular station, located at high altitude in a region rarely subject to significant influence of pollution from continental origin. The specific time period between 06:00 and 11:00 LT was chosen because it includes the usual nucleation hours at the site (see Sect. 3.1.2 and Foucart et al., 2018), as the main purpose of the present work is to evaluate the effect of volcanic plume conditions on NPF. This classification slightly differs from that introduced earlier by Foucart et al. (2018), who did not restrict the plume detection to morning hours, but focussed instead on daytime values, i.e. when global radiation was > 50 W m$^{-2}$, and required only one hourly average of the SO$_2$ mixing ratio ≥1 ppb to assess the occurrence of plume conditions. Also, in order to avoid any misclassification of the days, all data between 4[th] and 17[th] of February were excluded from the analysis, since the lack of SO$_2$ measurement prevented from a proper identification of the plume conditions at Maïdo during this eruptive period. Consequently, only the last three eruptions of the year 2015, which occurred in May and over the period August-October, are discussed in the present work.

In total, 30 days were classified as "plume days" following the abovementioned criteria, among which 1 was excluded from further analysis due to DMPS malfunctioning. All these 29 days were previously classified as plume days by Foucart et al. (2018), who identified in total 44 plume days with available DMPS measurement. The difference in the classifications arises

from the different time windows investigated in the two studies, i.e. morning nucleation hours (this study) vs daytime (Foucart et al., 2018), and from the criterion on the number of hourly averages of the SO2 mixing ratio ≥1 ppb needed for the plume detection. In fact, plume conditions were detected after 11:00 LT on 8 of the 15 additional plume days reported by Foucart et al. (2018), and the plume conditions lasted during only 1 or 2 hours on the remaining days. The majority of the 29 plume days (22/29) were identified during the longest of the eruptions of the Piton de la Fournaise observed in 2015, which occurred between end of August and late October. Six of the remaining plume days were detected in May, and the last day was identified during the very short eruption observed in late July – early August. In the end, after filtering the data for instrument malfunctioning and / or absence of measurements (71 days in total), 29 plume days and 250 non-plume days were included in the analysis. The 15 remaining days, with late or short plume occurrence, will only be shortly discussed in Section 3.1.1.

In addition to the abovementioned classification, we further analysed the characteristics of the plume days in terms of 1) the duration of the plume conditions detected at Maïdo (from 3 to 5 hours between 06:00 and 11:00 LT) and 2) the level of the hourly average $SO_2$ mixing ratios measured during the same time period. The plume was detected during the 5 hours of the time window of interest on 20 of the 29 days, and during 4 and 3 hours on 5 and 4 days, respectively. Concerning the $SO_2$ levels, they varied significantly from day to day. As an indicator of the "strength" of the plume, we calculated for each plume day the median of the $SO_2$ hourly averages ≥1 ppb identified between 06:00 and 11:00 LT. For 6 of the 29 plume days, the median $SO_2$ level was found between 1 and 2 ppb, and 17 days showed median levels relatively homogeneously distributed in the range between 2 and 14 ppb. Higher median $SO_2$ mixing ratios were observed on the remaining days, up to 249 ppb on the 20[th] of May. Based on this analysis, and in order to evaluate more specifically the effect of "strong" plume conditions on NPF (regarding both the duration of the plume conditions and the magnitude of the $SO_2$ levels), we further defined a sub-class of plume days, hereafter referred to as "strong plume days". The strong plume days were defined as plume days for which 1) the plume conditions lasted from 06:00 to 11:00 LT, with 2) the median of the $SO_2$ hourly averages ≥ 5 ppb. In total, 14 days were classified as strong plume days based on these criteria. Note that these days were included in the statistics reported for plume days in the next sections, and were in addition highlighted separately. The threshold of 5 ppb was chosen as it led to a fair compromise between the number of days to be classified as strong plume days, which we wanted to keep significant for the relevance of our analysis, and a mixing ratio of $SO_2$ significantly higher than that measured during non-eruptive periods. For comparison, 5 ppb corresponds to intermediate mixing ratios reported for polluted megacities (Mallik et al., 2014). As an illustration, Fig 1.b and c present the negative ion and particle number size distributions measured on two NPF event days detected during the eruptive period in May, in regular (May 29[th]) and strong (May 21[st]) plume conditions, respectively.

## 2.4 Sulfuric acid concentration in the gas phase

In the absence of direct measurements, we used a proxy to estimate the concentration of gaseous sulfuric acid. To our knowledge, there is no specific proxy dedicated to the rather unusual volcanic plume conditions, so we considered instead the expressions from Petäjä et al. (2009) and Mikkonen et al. (2011), which have already been widely used in nucleation studies. The two proxies have the common feature to consider the oxidation of $SO_2$ by OH as the only source of $H_2SO_4$, and do not

include the contribution of other oxidants possibly emitted together with $SO_2$ in the volcanic plume, and possibly prone to contribute to $H_2SO_4$ production, as discussed earlier by Berresheim et al. (2014) for the coastal atmosphere. The two proxies, however, differ in their construction, as Petäjä et al. (2009) used data collected solely in the boreal forest, while Mikkonen et al. (2011) used measurements from different locations, including the urban area of Atlanta, where, to a certain extent, $SO_2$ mixing ratios are resembling those measured at Maïdo in plume conditions. Besides the variety of measurement conditions, the expression from Mikkonen et al. (2011) also considers more parameters, including the temperature dependant reaction rate between $SO_2$ and OH as well as the relative humidity, which can fluctuate a lot and reach high values at mountain sites such as Maïdo. Given the above, the proxy developed by Mikkonen et al. (2011) was finally used in the present work. However, since the relevance of this proxy could not be evaluated in volcanic eruption plume conditions, neither from available measurements nor existing literature, one should keep in mind the potential limits of using such parametrization when interpreting the related results.

## 3. Results

### 3.1 NPF analysis in the volcanic plume

#### 3.1.1 Frequency of occurrence

As mentioned in Sect. 2.3, 29 plume days were identified at Maïdo as a consequence of the 3 eruptions of the Piton de la Fournaise which could be documented in 2015. Besides the plume days, 250 days with no influence of the volcanic plume were identified and included in the analysis (days with plume conditions detected in the afternoon or short plume occurrence were excluded, 15 days in total, see Sect. 2.3 for more details); these days will be hereafter referred to as "non-plume days". Figure 2 shows the monthly NPF frequency separately for plume and non-plume days, with a specific focus on May, August, September and October 2015, when the eruptions were observed. Note that statistics for non-plume days were previously reported for all months in 2015 by Foucart et al. (2018). Our results suggest that volcanic plume conditions favour the occurrence of NPF at Maïdo since all the plume days were classified as NPF event days with the exception of 3 days classified as undefined in September, leading to higher NPF frequencies in plume conditions compared to non-plume days over the months highlighted on Fig. 2 (90% vs 71%). Focussing on the strong plume days, 12 were classified as event days and the remaining 2 days were classified as undefined. At the annual scale, i.e. when including all months in the calculation, the NPF frequency was raised from the already remarkably high value of 67% when excluding the plume days to 69% when considering both plume and non-plume days. Such values are among the highest in the literature, similar to those previously reported for the high-altitude station of Chacaltaya (5240 m a.s.l., Bolivia) (64%, Rose et al., 2015) and the South African savannah (69%, Vakkari et al., 2011), slightly lower compared to that reported for the South African plateau, where NPF events are observed on 86% of the days (Hirsikko et al., 2012).

In addition, a quick analysis was also performed on the 8 days for which the volcanic plume was detected after the morning hours during which nucleation is usually initiated (see Sect. 3.1.2 for more details about the timing of NPF). With the exception of one day classified as undefined in October, all other days were classified as NPF event days, but there was no clear evidence of an effect of the "late" plume conditions on the ongoing events, triggered earlier during the day. High $SO_2$ mixing ratios (up to several hundreds of ppb) associated to plume conditions were also measured during the night, but they were not accompanied by any obvious particle formation nor growth process at Maïdo, supporting a determinant role of photochemistry in these secondary particle formation processes.

### 3.1.2 Timing of the events

As a first investigation of the specificities of NPF in plume conditions, we performed a simple analysis of the starting time of the NPF events on plume and non-plume days. The starting time of an event was defined by a visual inspection as the time when the 1.5-2.5 nm ions concentration measured with the AIS significantly increased. Only the events simultaneously detected with the AIS and the DMPS were included in this analysis, and the dataset was not limited to May-August-September-October, but included instead all available AIS data between mid-May and end of October. In total, 36 events observed on non-plume days and 10 events detected in plume conditions were documented.

The median starting time of NPF in non-plume conditions was found at 08:36 LT [25th percentile: 08:15 LT; 75th percentile: 09:06 LT]. Earlier rising time of the cluster concentration was observed on plume days, around 07:41 LT [07:18 LT; 08:16 LT]. In addition, we also calculated the median time laps between sunrise and beginning of NPF, since the starting time of NPF was most likely affected by the change in sunrise time over the course of the investigated period. Median time laps between sunrise and rising time of the cluster concentration was 2 hours and 11 minutes [1 hour and 55 minutes; 2 hours and 26 minutes] on non-plume days, with a minimum of 54 min observed on May 18th, and was about 45 minutes shorter in plume conditions, being 1 hour and 26 minutes [57 minutes; 1 hour and 38 minutes], with a minimum of 29 min obtained on August 28th. These observations suggest that on plume days, when precursors related to volcanic plume conditions were available prior to sunrise in a sufficient amount, photochemistry was the limiting factor for NPF to be triggered. In contrast, on non-plume days, NPF was certainly limited by the availability of condensable species, which were most likely transported from lower altitude by the mean of convective processes taking place after sunrise. Further discussion on the precursors involved in the process is reported in Sect. 3.2.2.

### 3.1.3 Particle formation and growth rates

Figure 3 shows the formation rate of 2 and 12 nm particles ($J_2$ and $J_{12}$, respectively), as well as the particle growth rate between 12 and 19 nm ($GR_{12-19}$). Note that statistics reported for plume days do include data from the strong plume days, which are in addition highlighted separately. Besides the high values observed on some of the strong plume days, which exceed those measured on regular plume days, $GR_{12-19}$ showed an important variability, as reflected by the monthly inter-quartile ranges, which were on average of the order of 80% of the corresponding medians. Also, with the exception of May, the monthly

medians of $GR_{12-19}$ were higher on non-plume days compared to plume days (Fig. 3.a). The overall effect of the plume conditions on the particle growth thus appeared to be limited. This observation is most likely related to the fact that not only the amount of precursor vapours (including for instance $SO_2$, see Fig. 1.a) was increased in the volcanic plume, but also the number concentration of the particles to grow, from both primary and secondary origin, as also reflected in the variations of the CS (Sect. 3.2.1) and discussed later in Sect. 3.3.1.

In contrast, the effect of plume conditions was more pronounced on the particle formation rates, both for $J_2$ and $J_{12}$ (Fig. 3.b and c). Indeed, with the exception of September, when slightly lower median values were found in plume conditions, the particle formation rates were on average increased on plume days, with the most important difference observed in May, when median $J_2$ and $J_{12}$ were increased by a factor of ~ 17 and 7.5, respectively, in plume conditions. However, most of the values obtained on strong plume days were similar to those measured during regular plume days of the same month. Higher particle formation rates observed on plume days indicate that particles were constantly formed in the volcanic plume along the transport pathway to Maïdo, showing that nucleation and growth taking place over a distance of the order of 40 km appears like a regional scale homogeneous process, which can be described with the usual equations (Eq. 1-4) recalled in Sect. 2.2. A rough estimate for the transport time of the particles nucleated in the vicinity of the volcano to the Maïdo observatory can be obtained by dividing the distance between the sites by the median wind speed measured on NPF event days: 39 km ÷ 1.8 m s$^{-1}$ ≈ 6 hours. This indicates that in such conditions, the $GR_{12-19}$ reported on Fig. 3.a were often sufficient (> 8 nm h$^{-1}$) for the newly formed particles (~ 1 nm) to grow up to CCN relevant sizes (~ 50 nm, see Sect. 3.3.2) during their transport, further explaining the observation of the typical banana shape of the events, both on plume and non-plume days. Similar analysis was repeated with the 75[th] percentile of the wind speed measured on NPF event days (2.9 m s$^{-1}$), and, again, the observed growth was often fast enough (> 13 nm h$^{-1}$) for the particles to reach 50 nm during the corresponding ~ 3 hours 45 minutes trip to Maïdo.

These observations suggest that, despite a limited effect on particle growth, plume conditions do affect NPF, both in terms of frequency of occurrence and particle formation rate. However, assessing the real effect of these specific conditions on the particle formation and growth is challenging. In specific, as previously highlighted in the companion study by Foucart et al. (2018), the particle growth rates calculated from high altitude stations such as Maïdo are "apparent" due to the complex atmospheric dynamics around these sites, and may in particular be overestimated due to the concurrent transport of growing particles to the site. Influence of the volcanic plume on larger particles, including CCN relevant sizes, is further investigated in Sect. 3.3., while the next section is focussed on the analysis of key atmospheric components previously reported to influence NPF, both in terms of frequency of occurrence and characteristics.

## 3.2 Effect of key atmospheric components on the NPF process

NPF has been previously reported to be influenced by various atmospheric parameters, including solar radiation, temperature (Dada et al., 2017), as well as RH, which effect on the process is certainly the less evident to predict and understand (e.g. Birmili et al., 2003; Duplissy et al., 2016). In the frame of the present analysis, the median diurnal variations of the

abovementioned parameters reported on Fig. S1 (in the Supplementary) did not highlight any specificity for the events observed on plume days, and displayed similar behaviour in and off-plume conditions.

In addition to the aforementioned meteorological variables, which are thought to influence directly the production of the nucleating and growing vapours as well as the survival of the newly formed clusters, other factors were shown to affect NPF, such as for instance the loss rate of the condensing compounds on pre-existing particles. The effect of the volcanic plume on this last parameter is discussed below, while the role of $H_2SO_4$, expected to play a determining role in the events observed on plume days (Boulon et al., 2011; Sahyoun et al., 2019), is investigated in the following section.

### 3.2.1 Condensation sink

As recalled in Sect. 2.1, the condensation sink represents the loss rate of precursor vapours on pre-existing larger particles, and is thus expected to affect directly the amount of precursors available for NPF. In order to further investigate the effect of this parameter on the occurrence of NPF and avoid any interference with the CS increase caused by the newly formed particles themselves, we focus here on the CS observed prior to usual nucleation hours, between 05:00 and 07:00 LT.

Figure 4.a. shows the monthly median of the CS calculated over the abovementioned time period, separately for plume, strong plume and non-plume days, event and non-event days. Note that strong plume days were included in the statistics reported for plume days, and were also highlighted separately. The comparison of non-plume NPF event and non-event days did not highlight any clear tendency over the months of interest for this study. Indeed, comparable median CS were observed in August regardless the occurrence of NPF later during the day, higher values were in contrast obtained on event days in May, while the opposite was observed in September and October, most likely related to biomass burning activity in South Africa and Madagascar during austral spring (Clain et al., 2009; Duflot et al., 2010; Vigouroux et al., 2012). The overall number of non-event days included in the statistics was, however, limited, and using comparable time window Foucart et al. (2018) reported that CS was on average higher on NPF event days compared to non-event days when including all the data from 2015. These contrasting results are representative of the observations from high altitude observatories at a larger scale, where both the location of the sites, their topography and the fast changing conditions related to complex atmospheric dynamics are likely to influence the effect of the CS on the occurrence of NPF. Indeed, Boulon et al. (2010) and Rose et al. (2015) reported that CS was on average positively correlated with the occurrence of NPF at Jungfraujoch (3580 m a.s.l., Switzerland) and Chacaltaya, respectively. These observations suggest that the availability of the precursors was often limiting the process at these sites, which seemed to be fed with vapours transported together with pre-existing particles contributing to the CS. In contrast, the CS was observed to be on average lower on NPF event days compared to non-event days at puy de Dôme, which could be in a less precursor-limited environment due to its lower altitude (Boulon et al., 2011).

Despite an important variability of the reported values, the median CS obtained on plume days were on average higher than those observed on non-plume days, with the largest difference in May. The CS reported for strong plume days were even higher, with median values on average increased by one order of magnitude compared to non-plume event days (up to 30 times higher in May). One might have expected those enhanced CS to inhibit NPF at Maido, which was instead more frequent in

plume conditions compared to non-plume days (Sect. 3.1.1). This non-intuitive result is most likely explained by the increased mixing ratios of $SO_2$ (Fig. 4.b), and in turn $[H_2SO_4]$, which were concurrently measured on strong plume days. Assuming that $H_2SO_4$ was contributing to NPF in such conditions, as previously suggested by Boulon et al. (2011) and Sahyoun et al. (2019), increased $SO_2$ emissions probably compensated for the strengthened loss rate of the condensing vapours involved in particle

formation and growth on plume days, and eventually let NPF occur. Interestingly, comparable observations were recently reported by Hakala et al. (2019) from a rural background site in western Saudi-Arabia. There, NPF was observed in sulphur rich plumes of anthropogenic origin, and did not seem to be limited by the presence of pre-existing particles either; instead, the particle formation and growth rates were shown to increase as a function of the CS. Similarly, the balance between the amount of $SO_2$ and the magnitude of the CS most likely influenced the strength of the events observed at Maïdo, and explained

in specific the variable trends highlighted earlier in the comparison of the particle formation rates calculated on plume and non-plume days (see Sect. 3.1.3, Fig. 3.b and c). Indeed, as reported previously, the largest CS increase observed between non-plume and plume NPF event days occurred in May, when $SO_2$ mixing ratios were also the highest (Fig. 1), with a median of 26.7 ppb [25th percentile: 1.1 ppb; 75th percentile: 120.5 ppb] calculated during nucleation hours (06:00 and 11:00 LT). We may thus hypothesize that the resulting conditions were highly favourable to NPF, and not only lead to high NPF frequency

(Fig. 2), but also to stronger events, with increased particle formation rates compared to non-plume days (Fig. 3.b and c). In September and October, the median CS measured in plume conditions were comparable to that observed in May (Fig. 4.a), but $SO_2$ mixing ratios were in contrast lower during nucleation hours, with medians around 3.4 ppb [1.5 ppb; 5.6 ppb] and 3.8 ppb [1.9 ppb; 16.9 ppb], respectively. This most likely resulted in less favourable conditions for NPF than in May, which in turn did not enhance the particle formation rates compared to non-plume days. Higher CS observed on plume days also supported

the fact that in plume conditions, as suggested in the previous section, the number of particles to grow was increased compared to non-plume days, and the concurrent strengthening of the precursor source rate was on average not sufficient to result in faster particle growth. Nonetheless, while it was possible to evidence the abovementioned trends with our statistical approach, one should keep in mind that both the occurrence and characteristics of NPF are likely to be affected at very short time scales due to the variable nature of the volcanic eruptions. Deeper investigation of the effect of the volcanic eruption plume on NPF

would thus require more detailed analysis of the event to event variability, which was, however, beyond the scope of the present work.

The origin of the particles responsible for increased CS prior to nucleation hours on plume event days remains uncertain, but the high $SO_2$ mixing ratios which were measured concurrently suggest a volcanic origin for these accumulation mode particles (see Sect. 3.3.1 for more details about the shape of the particle number size distribution). The presence of accumulation mode

particles emitted as primary ash have seldomly been reported in the literature, as the instrumentation used for volcanic studies is not adapted for measuring such small sizes. In the case of the Piton de la Fournaise, the existence of such particles was, however, very unlikely. Indeed, this basaltic volcano was reported to only emit negligible amount of ash, which were observed in the form of Pele's hairs during very specific eruptions (Di Muro et al., 2015); if present, we would have expected the fragments of ash to also cause an increase of the particle mass in the coarse mode, which was not obvious during the eruptions

observed in 2015 (Tulet et al., 2017). Other nocturnal sources of aerosols at the vent of the volcano and transport of these particles to Maïdo were in contrast more probable. A first possible production pathway is related to the fact that the Piton de la Fournaise is characterized by an usual water loading, reflected by high $H_2O/SO_2$ ratios (Tulet et al., 2017). In specific, there is continuous formation of liquid water at the vent of the volcano; gaseous $SO_2$ can thus be dissolved into the droplets, leading

to aqueous formation of $H_2SO_4$, which likely further condense onto pre-existing particles after evaporation of the cloud in the vicinity of the Piton de la Fournaise (Tulet et al., 2017). Catalytic oxidation of $SO_2$ inside the volcanic dome was also reported as a potential source of $H_2SO_4$ by Zelenski et al. (2015) based on measurements conducted at Bezymianny volcano. Finally, Roberts et al. (2019) recently suggested high temperature chemistry as a source of $H_2SO_4$ precursors in the near-source volcanic plume using model simulations. The abovementioned particles, originally formed or transformed *via* heterogeneous

mechanisms occurring in the very close vicinity of the vent, will be, despite the transformations they probably further undergo during their transport, referred to as volcanic primary particles hereafter, as opposed to secondary particles resulting from gas-to-particle conversion processes that take place further from the volcanic dome. This nomenclature is consistent with earlier results from Allen et al. (2002), who reported the presence of primary sulphate aerosols at Masaya volcano. These so-called primary particles are also likely to participate in daytime particle concentration, together with photochemically-driven

secondary formation pathways.

### 3.2.2 The role of sulfuric acid

As previously mentioned in Sect. 2.4, sulfuric acid concentrations were obtained from $SO_2$ mixing ratios using the proxy by Mikkonen et al. (2011). $H_2SO_4$ has often been reported to play a key role in the early stage of atmospheric NPF (Sipilä et al., 2010; Kulmala et al., 2013; Yan et al. 2018), and in particular in sulphur-rich volcanic plume conditions (Mauldin et al., 2003;

Boulon et al., 2011; Sahyoun et al., 2019), thus motivating our specific interest in this study.

Figure 5.a shows all $J_2$ derived during the NPF events detected in plume conditions as a function of $[H_2SO_4]$. Highest $[H_2SO_4]$ were mostly observed on strong plume days and coincided with the highest $J_2$, but overall, the relationship between $J_2$ and $[H_2SO_4]$ did not appear to be significantly different on strong plume days compared to regular plume conditions. In order to further investigate the connection between the cluster particle formation rate and the abundance of $H_2SO_4$, $J_2$ was fitted with a

simple power model $J_2 = k \times [H_2SO_4]^a$, in a similar way as previously done in earlier studies (Kulmala et al., 2006; Sihto et al., 2006; Kuang et al., 2008; Sahyoun et al., 2019). When considering all data in the fitting procedure, i.e. including strong plume days, parameters $k$ and $a$ were found to be $1.14 \times 10^{-10}$ and 1.16, respectively. The correlation between $J_2$ and $[H_2SO_4]$ was moderate ($R^2 = 0.23$), but significant, as indicated by the corresponding *p*-value (p = $2.35 \times 10^{-24}$). As a reminder, the *p*-value is commonly used to quantify the statistical significance, and indicates in the context of this study the maximum

probability for the correlation observed between $J_2$ and $[H_2SO_4]$ to result from a coincidence.

The values calculated for $k$ and $a$ were slightly different from those reported by Kuang et al. (2008) for a set of 7 stations, including 2 insular marine sites in the Pacific Ocean and 5 urban or rural continental sites located in Northern America and Europe, where the authors systematically found $a$ in the range 1.98-2.04 and $k$ between $\sim 10^{-14}$ and $10^{-11}$. These values were,

however, calculated from a limited number of events for each site (from 1 to 9 events per site), and were derived from the correlation between $J_1$ (instead of $J_2$ in the present work) and measured [$H_2SO_4$] (instead of proxy-derived in ours study). Also, none of the NPF events investigated by Kuang et al. (2008) occurred in volcanic plume conditions. Overall, the values derived on plume days at Maïdo were in contrast more consistent with those recently reported by Sahyoun et al. (2019) based on aircraft measurements conducted in the passive plume of Etna ($k = 1.84 \times 10^{-8}$, $a = 1.12$), despite slight differences in the analysis. Indeed, Sahyoun et al. (2019) used direct measurements of [$H_2SO_4$] and investigated the link between [$H_2SO_4$] and $J_{2.5}$ instead of $J_2$. Also, instead of fitting all derived $J_{2.5}$, the dataset was first binned with respect to [$H_2SO_4$] (bins of equal length, $0.25 \times 10^8$ cm$^{-3}$), after which median $J_{2.5}$ were calculated for each bin and finally fitted using a power model. As a sensitivity study, and in order to provide a more consistent comparison with the results reported by Sahyoun et al. (2019), we applied similar fitting method; only difference was in the binning procedure, as we used bins with an equal number of data points instead of equal length. The results of this analysis are shown on Fig. S2, in the Supplementary. Applying the fit on median $J_2$ led to comparable fitting parameters ($k = 4.36 \times 10^{-10}$, $a = 1.09$), but the resulting correlation between $J_2$ and [$H_2SO_4$] appeared to be significantly stronger (R² = 0.88, p = $5.59 \times 10^{-5}$). We also investigated the impact of the number of bins on the final parameters and goodness of the fit (Table S1 in the Supplementary), but varying the number of bins between 10 and 30 did not led to major differences.

In order to get further insight into the nucleation mechanism likely to explain the observed events, we additionally compared the formation rates derived from DMPS measurements with that predicted by the recent parameterization developed by Määttänen et al. (2018), which describes neutral and ion-induced binary nucleation of $H_2SO_4$-$H_2O$. We used the Fortran code included in the supplementary electronic material of the paper, and, in a first approach, we run the model using the average of the temperature, relative humidity and CS calculated during nucleation hours on plume event days. We used an average value of 3 cm$^{-3}$ s$^{-1}$ for the ion pair production rate, consistent with previously reported values from different sites, including the high-altitude station of puy de Dôme (Rose et al. 2013 and references therein). Otherwise, all settings and parameters were those set by default in the Fortran code. According to model results, the total formation rate of 2 nm-clusters could be mostly explained by ion induced nucleation for [$H_2SO_4$] below ~ $8 \times 10^8$ cm$^{-3}$, while neutral pathways might contribute to a larger extent at larger sulfuric acid concentrations (although we do not have enough data points at these higher sulphuric acid levels to provide a statistically robust conclusion). The measurement of 2 nm-ions formation rates would have certainly helped confirming the major role of ions in volcanic plume. However, in absence of (neutral) particle measurements below 3 nm, which are needed to evaluate the attachment of ions on neutral clusters (Kulmala et al., 2012), we could not derive ion formation rates from observational data. While the abovementioned attachment term has been shown to contribute little to the formation of 2-nm ions in some environments and / or conditions, where it could in turn be neglected (eg: Buenrostro Mazon et al., 2016), the lack of knowledge in volcanic plume conditions prevented from the use of such assumption in the present work.

A deeper analysis of the ability of the model to represent the observed events was then performed, and for that purpose the model by Määttänen et al. (2018) was run for each NPF event, with the corresponding temperature, relative humidity and CS levels. Figure 5.b shows, for the same dataset as in Fig. 5.a, the ratio between $J_2$ derived from DMPS measurements (referred

to as $J_{meas}$) and $J_2$ predicted by the parameterization (referred to as $J_{param}$), which included both neutral and ion-induced nucleation pathways. In addition, we also binned the data with respect to [$H_2SO_4$] (10 bins with equal number of points) and calculated the median ratio $J_{meas}/J_{param}$ in each bin. As evidenced on Fig. 5.b, 64% of the calculated ratios were < 1, which was unexpected since particle formation rates resulting from the binary nucleation of $H_2SO_4$-$H_2O$ should not exceed $J_{meas}$.

This result was most likely related to 1) the use of proxy-derived [$H_2SO_4$] in the calculation of $J_{param}$ and 2) the fact that $J_{meas}$ was not derived from direct measurement of 2 nm particles but from $J_{12}$ and $GR_{12-19}$, which might have led to additional uncertainty. However, despite this inconsistency, Fig. 5.b illustrates our ability to predict $J_2$ from $SO_2$ mixing levels with a relatively fair accuracy, especially for [$H_2SO_4$] in the range between $1.5 \times 10^8$ and $7 \times 10^8$ cm$^{-3}$. Indeed, the medians of $J_{meas}/J_{param}$ highlighted on Fig. 5.b indicates that $J_{param}$ was on average within a factor of ~ 3 of $J_{meas}$ for [$H_2SO_4$] < $10^9$

cm$^{-3}$. Higher discrepancies were in contrast observed for [$H_2SO_4$] > $10^9$ cm$^{-3}$, which may stem from a reduced predictive ability of the proxy by Mikkonen et al. (2011) for the highest $SO_2$ mixing ratios, since such values were not used in the construction of the proxy, or from the binary nucleation parameterization not being fully adapted to the volcanic plume environment. Nonetheless, despite the abovementioned limits, we believe that the last results are of high interest, because they show that in absence of direct measurement of [$H_2SO_4$] and sub-3 nm particles concentration, the knowledge of $SO_2$ mixing ratios can lead

to a fair approximation of $J_2$.

All together, these results suggest that higher $SO_2$ mixing ratios observed on plume days did contribute to the NPF events observed in such conditions, in agreement with earlier results from Boulon et al. (2011) and Sahyoun et al. (2019). However, the contribution of other compounds to the process could not be excluded based on the available dataset, and additional measurements would be needed to further investigate this aspect, including in specific direct measurement of the chemical

composition of the clusters as well as their precursors. Such measurements would also allow more detailed evaluation of the proxy by Mikkonen et al. (2011) for the prediction of [$H_2SO_4$] in volcanic eruption plume conditions. Together with direct measurements of sub-3 nm cluster ions and particles, the identification of the precursors would finally favour a more robust investigation of the actual mechanisms responsible for cluster formation in volcanic plumes.

### 3.3 Particle growth up to climate relevant sizes in the volcanic plume

The previous section was dedicated to the analysis of NPF occurrence and characteristics in volcanic plume conditions. In this section, we further investigate the effect of such process on the shape of the particle size distribution, first including all sizes between 10 and 600 nm, and then focussing more specifically on large-enough particles to act as CCN. Since the results we have reported so far only revealed limited signature of strong plume conditions on NPF characteristics, we will no longer put any specific focus on these particular days in this last section. They will, however, still be included in the statistics reported

for plume days.

### 3.3.1 General features of aerosol particle size distributions

The effect of NPF and/or plume conditions on the particle number size distribution was investigated based on the hourly median particle spectra measured with the DMPS in different conditions between 07:00 and 16:00 LT (Fig. 6). This time period was selected as, besides usual NPF hours, it also includes one hour prior to nucleation hours, which allowed to study the main features of the particle size distribution in the different conditions (plume and non-plume) without fresh influence of NPF, as well as several hours to investigate the change of the spectra caused by particle growth processes. Note that in order to increase the statistical relevance of the results (especially for non-event days), the analysis was not restricted to May-August-September-October, and all available data was included in the analysis. All median spectra were in addition fitted with four Gaussian modes, including nucleation, Aitken and 2 accumulation modes, which parameters are shown on Fig. 7 and also reported in Table A1 of the Appendix.

As previously mentioned, the median spectra recorded at 07:00 LT, i.e. prior to nucleation hours, gave a unique opportunity to compare the main features of the particle number size distributions recorded in plume and non-plume conditions in absence of freshly nucleated particles. The spectra measured on non-plume days (both event and non-event days) displayed comparable shapes as well as similar concentrations, while higher concentrations were in contrast measured in plume conditions. As discussed in Section 3.2.1, these differences, which were the most pronounced for the 2 accumulation modes, were most likely explained by the presence of particles originating from heterogeneous processes occurring at high temperatures at the vent during the eruptive periods, and assimilated to volcanic primary particles. Indeed, in plume conditions the population of the first accumulation mode (modal diameter ~ 100 nm) was on average around 300 $cm^{-3}$, against 55-80 $cm^{-3}$ on non-plume days, indicating the presence of 220-245 $cm^{-3}$ additional primary particles originating from the volcano on plume days. Following similar reasoning, the contribution of volcanic primary particles to the concentration of the second accumulation mode (modal diameter ~ 190 nm) was around 118-132 $cm^{-3}$. Hence, there was on average almost a six-fold increase in the accumulation modes particle number concentration due to the emission of primary volcanic particles. Regarding the diameters of the modes, they did not appear to be significantly affected by the different atmospheric conditions.

Despite some variations of the particle concentration in the different modes, the shape of the spectrum observed at 07:00 LT remained the same throughout the investigated time window on non-event days. The concentrations of the Aitken and first accumulation modes were increased by a factor of ~ 2 between 07:00 LT and 13:00 LT, from 130 to 310 $cm^{-3}$ and 55 to 112 $cm^{-3}$, respectively, and concurrently the concentration of the second accumulation mode was multiplied by ~ 4, from 32 to 142 $cm^{-3}$, probably due to the transport of pre-existing particles from lower altitude. Surprisingly, the most important variation was observed for the concentration of the nucleation mode, with a 5-fold increase from 18 to 86 $cm^{-3}$ between 07:00 LT and 12:00 LT despite the absence of NPF. These concentrations were, however, significantly lower compared to those observed on NPF event days, up to 5700 $cm^{-3}$ in plume conditions. Also, in contrast with NPF event days, the diameters of the modes, including in specific that of the nucleation mode, remained stable throughout the investigated time window (modal diameter ~ 15nm, ~

36 nm, ~ 83 nm and ~ 164 nm for the nucleation, Aitken, first and second accumulation modes, respectively), indicating the absence of a growth process characteristic of NPF.

Consistent with previous observations reported in Sect. 3.1.2, the starting of NPF was seen at 08:00 LT on event days (regardless the occurrence of plume conditions) from a visual analysis of the spectra, and was further confirmed by the increase of the particle concentration in the nucleation mode, which lasted until 11:00 LT. The most significant change was observed in plume conditions, with a thousand-fold increase of the nucleation mode particle concentration in 4 hours, from 5 to 5700 $cm^{-3}$, i.e. two orders of magnitude stronger than that observed on non-plume days (from 25 to 1700 $cm^{-3}$), consistent with the higher particle formation rates observed on plume days (see Sect. 3.1.3). After 11:00 LT, the particle concentration in the nucleation mode was observed to decrease, down to 300 $cm^{-3}$ and 990 $cm^{-3}$ at 16:00 LT, on plume and non-plume days, respectively, most likely because of the growth of particles outside of the mode, as well as their loss on larger particles through coagulation processes. Concurrent increase of the modal diameter was also observed between 07:00 and 13:00 LT on plume days, from 12 to 26 nm, and slightly later, between 09:00 and 16:00 LT, on non-plume days, from 10 to 30 nm, further illustrating the simultaneous formation and growth of the particles.

Still focussing on NPF event days, the changes observed in the parameters of the Aitken and 2 accumulation modes were less pronounced than for the nucleation mode. With the exception of the slight difference observed at 07:00 LT, the Aitken mode displayed similar diameters on plume and non-plume days, with only limited variations over the investigated time window, especially on non-plume days (32-45 nm and 38-46.5 nm, on plume and non-plume days, respectively). The initial concentration of the Aitken mode measured in plume conditions was slightly higher than on non-plume days (270 vs 180 $cm^{-3}$), indicating that, in addition to accumulation mode particle sizes, primary particles as small as ~ 40 nm might have been produced by the volcano. The increase of the particle concentration observed until 13:00-14:00 LT was also more pronounced on plume days (~ 7-fold increase in plume conditions, up to 1800 $cm^{-3}$, vs ~ 4-fold increase on non-plume days, up to 750 $cm^{-3}$), indicating the presence of ~ 1050 $cm^{-3}$ additional particles in the Aitken mode due to the presence of the plume. This observation was consistent with the enhanced production of particles previously reported for lower sizes in plume conditions, as the particles in the Aitken mode most likely resulted from the growth of smaller particles originating from the nucleation mode. As already mentioned, the concentrations of the 2 accumulation modes measured at 07:00 LT were both significantly higher on plume days (300 and 150 $cm^{-3}$, for the first and second accumulation mode, respectively) compared to non-plume days (80 and 18 $cm^{-3}$, respectively), most likely due to additional sources of particles at the vent of the volcano during eruptive periods (see Sect. 3.2.1). Later on, the transport of primary particles originating from the urban areas located at lower altitude as well as the growth of the newly formed particles most likely contributed to the concentration increase observed for the accumulation modes during the course of the day, up to 770 and 690 $cm^{-3}$, respectively, on plume days, against 445 and 130 $cm^{-3}$ on non-plume days. With the hypothesis that vertical transport from the boundary layer was the same on plume and non-plume days, we could evaluate that the volcanic plume secondary aerosol formation contributed to ~ 325 $cm^{-3}$ and ~ 560 $cm^{-3}$ in the first and second accumulation modes, respectively. Concerning the diameter of the modes, that of the second accumulation mode showed limited variations on plume days (185-200 nm), being closed to that observed on non-plume days

(191 nm). In contrast, the initial diameter of the first accumulation mode was higher on non-plume days (97 vs 81.5 nm) and also showed a more pronounced increase up to 146 nm over the investigated time period.

Altogether, one can infer from these measurements a distribution of the particles of volcanic origin, including the contributions of both primary and secondary aerosols. Following the above analysis, the concentration of the so called volcanic primary particles was calculated for each mode as the difference between the concentrations measured at 07:00 LT on plume and non-plume days, and the values obtained at 07:00 LT in plume conditions were used for the other characteristics of the modes (i.e. sigma and modal diameter). In addition, the concentration of secondary aerosol particles of volcanic origin was calculated for each mode as the difference between the maximum concentrations observed on plume event days and that observed on non-plume event days, which were found between 11:00 and 14:00 LT depending on the modes and conditions; the values obtained at 12:00 LT in plume conditions, when the effect of secondary aerosol formation on the spectrum was on average the most pronounced, were used for the other characteristics of the modes. Resulting aerosol spectrum is reported on Fig. 8, with the detailed contributions of volcanic primary and secondary particles for each mode. Secondary aerosol particles formed due to the presence of the plume contributed 93% of the total concentration observed on plume event days, clearly dominating all the modes but the first accumulation mode, for which the contribution of volcanic primary particles was more significant. The presence of a secondary contribution to the accumulation modes is likely the result of the growth of particles from the Aitken mode, due to the presence of more condensable gases.

### 3.3.2 Investigation of the formation of potential CCN during NPF events

The increase of potential CCN concentration during NPF was investigated using DMPS measurements, in a similar way as done earlier by Rose et al. (2017) for the high-altitude station of Chacaltaya, following the approach originally developed by Lihavainen et al. (2003). It is based on the hypothesis that the lower cloud droplet activation diameter $d_{act}$ of aerosol particles is in the range 50 – 150 nm for the typical supersaturations (SS) encountered in natural clouds, including those forming at high altitude (Jurányi et al., 2011; Hammer et al., 2014). For instance, direct CCN measurements conducted at the high altitude station of puy de Dôme with a dedicated chamber (Roberts and Nenes, 2005) showed that particles in the range 50-150 nm were activated at SS 0.24%, also reported to be representative of in-cloud SS at the site (Asmi et al., 2012). Following this basic assumption, the concentration of potential CCN can be assimilated to the particle concentration $N_{>d_{act}}$ measured above any given $d_{act}$ in the range 50 - 150 nm. Sensitivity studies are usually performed using multiple activation diameters, which reflect the effect of both the properties of the particle itself (such as the chemical composition) and atmospheric conditions (such as the supersaturation) on the ability of a particle to activate into a cloud droplet (e.g. Kerminen et al. 2012 and references therein). We used in a first approach the same activation diameters (50, 80 and 100 nm) as Rose et al. (2017), and we based our analysis on both the time series of the concentration of particles larger than these thresholds (hereafter referred to as $N_{50}$, $N_{80}$ and $N_{100}$, for 50, 80 and 100 nm, respectively) and the overall shape of the event reflected by the corresponding surface plot. This indirect method based on DMPS measurements only provides estimations of potential CCN concentrations instead

of real concentrations as measured by CCN chambers (Roberts and Nenes, 2005); however, for simplicity, we refer to these potential CCN as CCN hereafter.

This last analysis was not restricted to the months when the volcanic activity was detected, and 193 NPF event days identified in 2015 were included in the analysis (167 non-plume days and 26 plume days). As reported earlier by Foucart et al. (2018), the growth of particles > 80 – 100 nm was observed as a common feature of a large majority of the investigated days besides NPF, regardless the occurrence of plume conditions. In addition, high background concentrations were frequently seen above ~ 50 nm, most likely caused by the transport of pre-existing particles to the station. The influence of these phenomena on the variations of $N_{80}$ and $N_{100}$ was obvious, and often hindered the identification of a concurrent impact of NPF on the particle concentration increase at these sizes. In contrast, the background concentration was often significantly lower at 50 nm, which made it easier to unambiguously assess the growth of the newly formed particles up to 50 nm. Further evaluation of the contribution of NPF to the formation of CCN was thus finally performed using this single activation diameter, assuming $N_{50}$ was a good proxy for the concentration of particles likely to act CCN, or to become CCN after experiencing further growth.

The increase of $N_{50}$ observed during the events was attributed (at least partly) to NPF on 70 days, 15 of them being plume days and the remaining 55 being non-plume days. On the other 123 event days included in the analysis, the contribution of NPF to the increase of $N_{50}$ was more uncertain, but could not be excluded. Resulting frequencies of NPF contributing to the increase of > 50 nm CCN were thus 33% for non-plume days and 58% for plume days (Fig. 9.a). The increase of $N_{50}$ observed during NPF was in addition quantified: $N_{50-MAX}$, the maximum of $N_{50}$ observed during the event was compared to $N_{50-REF}$, calculated as the two-hour average of $N_{50}$ between 05:00 and 07:00 LT, prior to nucleation hours. Note that for a given NPF event, the identification of $N_{50-MAX}$ was limited to the time period 07:00 - 19:00 LT, because of the fast change of air masses and / or wind direction which is often observed at Maïdo after 19:00 LT (see Fig. S3 in the Supplementary). The absolute increase of $N_{50}$ was further calculated as the difference between $N_{50-MAX}$ and $N_{50-REF}$, and the median CCN productions observed in the different conditions are reported on Fig. 9.b. The median of the $N_{50}$ absolute increase observed on non-event days was around 420 cm$^{-3}$, and was significantly enhanced on event days, being around 1600 cm$^{-3}$ on non-plume days and 3720 cm$^{-3}$ in plume conditions. The more pronounced concentration increase observed on event days, and in specific in plume conditions, was explained by the multiplication of the sources of such particles on those specific days. Indeed, on non-event days, the variations of $N_{50}$ observed at Maïdo were probably caused exclusively by the transport of pre-existing large particles originating from the nearest urban areas. This process was itself tightly connected to the dynamics of the boundary layer and associated wind pattern, resulting in maximum concentrations around 13:00 LT (i.e. 09:00 UTC), as evidenced on Fig. 9.d. On event days, the diurnal variation of $N_{50}$ was strengthened due to the concurrent formation of secondary aerosols, i.e. including the formation and growth of new particles as well as the growth of pre-existing larger particles mentioned earlier (Fig. 9.d). Additional contribution of large particles formed *via* heterogeneous processes close to the vent of the volcano (and denoted as volcanic primary particles in the present work) was finally highly probable in plume conditions, as discussed in Sect. 3.2.1 and 3.3.1. This last hypothesis was further supported by the increased level of $N_{50}$ observed during the night on plume days compared to other days (Fig. 9.d).

In a similar way as done previously by Rose et al. (2017), we made an attempt to decouple the contributions of the abovementioned CCN sources on event days. For that purpose, the transport of pre-existing large particles from the boundary layer was first assumed to have similar magnitude on event and non-event days, regardless the occurrence of plume conditions. We also made the assumption that the contribution of volcanic primary particles did not vary significantly along the day in

plume conditions, and was thus systematically removed when calculating the difference between $N_{50-MAX}$ and $N_{50-REF}$. Following these hypotheses, the contribution of secondary aerosols to the observed CCN population was estimated from the difference between the median of the $N_{50}$ absolute increase observed on event days (i.e. resulting from transport of particles from the boundary layer and secondary aerosol formation) and that of non-event days (resulting from transport only). Resulting concentrations attributed to secondary aerosols were ~ 1180 cm$^{-3}$ and ~ 3300 cm$^{-3}$ for non-plume and plume event days,

respectively, and dominated the increase of $N_{50}$ observed on event days. From these average concentrations we could also infer the formation of secondary aerosols prone to act as CCN in plume conditions, with ~ 2120 cm$^{-3}$ additional particles detected on plume days compared to non-plume days. This stronger increase of the CCN concentration was most likely explained by the larger amount of condensable vapours available in plume conditions, including in specific sulfuric acid, as evidenced on Fig. 9.c. Indeed, the most significant $N_{50}$ increases coincided with high [$H_2SO_4$], suggesting that the growth of

more particles to CCN relevant sizes was favoured during NPF events occurring in the presence of large amount of $H_2SO_4$ caused by the eruptions. The increase of $N_{50}$ due to secondary aerosol formation in the presence of the plume was also consistent and of the same order of magnitude as the increase of the particle concentration reported in similar conditions for the Aitken and accumulation modes in the previous section.

## 4. Summary and conclusions

We investigated the occurrence of NPF in volcanic plume conditions at the Maïdo observatory based on measurements conducted between January 1$^{st}$ and December 31$^{st}$ 2015. During this time period, 4 effusive eruptions of the Piton de la Fournaise, located ~ 39 km away from the station, were observed, and we were able to detect volcanic plume conditions at Maïdo during 3 of the 4 eruptions. In total, 29 "plume days" were identified according to $SO_2$ mixing ratios measured during morning hours (06:00 – 11:00 LT, i.e. including usual nucleation hours), among which 14 days classified as "strong plume

days", and were compared with 250 "non-plume days". This dataset is, to our knowledge, the largest ever reported for the investigation of NPF in volcanic plume conditions, and allowed for the first time a statistical approach to characterize the process and also assess its relevance with respect to non-plume conditions.

Focussing on the months during which the volcanic plume was detected at Maïdo (May, August, September and October), NPF was observed on 90% of the plume days vs 71% of the non-plume days. On plume days, when higher amounts of

precursors (such as $SO_2$) were available prior to sunrise due to the occurrence of plume conditions, NPF seemed to be mainly limited by photochemistry, and was triggered early after sunrise. The process was in contrast observed later on non-plume days, most likely due to the lack of precursors before their transport from lower altitude after sunrise, as a result of convection.

With the exception of September, particle formation rates, both $J_{12}$ and $J_2$, were on average significantly increased on plume days. In contrast, despite high values of $GR_{12-19}$ reported on strong plume days, the overall effect of plume conditions on particle growth between 12 and 19 nm appeared to be limited, most likely because the number concentration of particles to grow was significantly raised in plume conditions, as reflected also by the larger CS observed on plume days.

Signature of the volcanic plume on the aerosol spectra up to 600 nm was further investigated based on the analysis and fitting of the particle size distributions recorded in the different conditions. The spectra measured prior to nucleation hours (07:00 LT) gave a unique opportunity to compare the main features of the particle number size distributions recorded in plume and non-plume conditions in absence of freshly nucleated particles. Main differences were observed for the two accumulation modes, which were more densely populated in plume conditions compared to non-plume days, most likely because of the

contribution of particles formed *via* heterogeneous processes at the vent of the volcano during eruptive periods, and assimilated to primary volcanic particles in the present work. The particle size distribution only experienced limited changes on non-event days, but significant variations of the particle concentration were in contrast observed for the nucleation and Aitken modes on NPF event days between 08:00 and ~11:00 LT, especially in plume conditions. An average size distribution of the particles of volcanic origin was further inferred from the measurements performed on event days in and off-plume conditions. The

contribution of secondary particles to the total concentration was around 93%, and clearly dominated that of primary particles for all but the first accumulation mode concentration, for which primary particles contributed significantly (40%).

Specific attention was further paid to the concentration of particles > 50 nm ($N_{50}$), assuming they could be used as a proxy for potential CCN population. The variations of $N_{50}$ were limited on non-event days, and attributed mainly to the vertical transport of pre-existing particles from the boundary layer, evaluated to ~ 420 cm$^{-3}$. The contribution of secondary particles to the

increase of $N_{50}$ was more frequent in plume than off-plume conditions, and the magnitude of the increase due to secondary aerosol formation was ~ 3300 cm$^{-3}$ compared to ~ 1180 cm$^{-3}$ on non-plume days.

In order to investigate deeper the influence of volcanic plume conditions on the occurrence of NPF and related effects on particle concentration, we first investigated the variations of several atmospheric parameters (temperature, relative humidity and global radiation). Similar patterns were observed in and off-plume conditions, without any specificity for the events

observed on plume days. Attention was then paid to the variations of the condensation sink (CS) and [H$_2$SO$_4$], previously reported to play a key role in the process. In order to avoid any interference with the CS increase caused by the newly formed particles themselves, we focussed on the CS slightly prior to nucleation hours. The comparison of non-plume NPF event and non-event days did not highlight any clear tendency over the months of interest for this study. In contrast, the median CS obtained on plume days were on average higher than those observed on non-plume days. Increased SO$_2$ mixing ratios measured

concurrently in plume conditions most likely compensated for the strengthened loss rate of the vapours and let NPF happen in the form of stronger events (with respect to $J_{12}$ and $J_2$) compared to non-plume days, suggesting at the same time a key role of H$_2$SO$_4$ in the process, consistent with recent observations in the plumes of Etna and Stromboli (Sahyoun et al., 2019). In order to test this last hypothesis, we derived [H$_2$SO$_4$] from SO$_2$ mixing ratios using a proxy available in the literature (Mikkonen et al., 2011). Despite a moderate strength, the correlation between $J_2$ and [H$_2$SO$_4$] was found to be significant. In addition, we

also evaluated the contribution of binary nucleation of $H_2SO_4 – H_2O$ to the observed events using the recent parameterization by Määttänen et al. (2018). Within the uncertainties associated to our calculations, we showed that in plume conditions, predicted $J_2$ from calculated $H_2SO_4$ concentration were fair approximations of $J_2$ derived from the measured $J_{12}$, indicating that $H_2SO_4$ was the main nucleating species in the plume. This result also highlighted the possibility to get relatively good

5    estimates of $J_2$ in absence of direct measurement of $[H_2SO_4]$ and sub-3 nm particles concentration, but only from the knowledge of $SO_2$ mixing ratios. The use of the parameterization finally gave insights into the pathways responsible for the formation of 2-nm cluster in plume conditions, and highlighted in specific the dominant role of ion-induced nucleation for $[H_2SO_4]$ below ~ $8\times10^8$ $cm^{-3}$, while neutral pathways were in contrast the most efficient above this threshold.

All together, our observations show that, based on one year of data, volcanic plume conditions favour the formation of particles

10   that frequently grow to CCN sizes within the first 40 km of the volcano's vent. The quantification of the contribution of primary vs secondary aerosol formation within a volcanic eruption plume on a statistical basis contributes to better understanding of this natural process, which might have contributed significantly to NPF and CCN formation in the pristine preindustrial era. Nonetheless, our study should be complemented in the future with a direct analysis of 1) the cluster formation rates, both charged and neutral and 2) the precursor vapours involved in their formation. Indeed, our approach to assess the role of $H_2SO_4$

15   was based on calculations, indirect measurements and model results, and we cannot exclude the contribution of other compounds, such as for instance halogens, and in particular iodine compounds, which were previously identified in volcanic plumes (Aiuppa et al., 2009) and also reported to contribute to NPF in coastal zones (Sipilä et al., 2016).

## 5. Data availability

DMPS data are accessible from the EBAS website (http://ebas.nilu.no/). AIS and $SO_2$ data can be provided upon request.

40

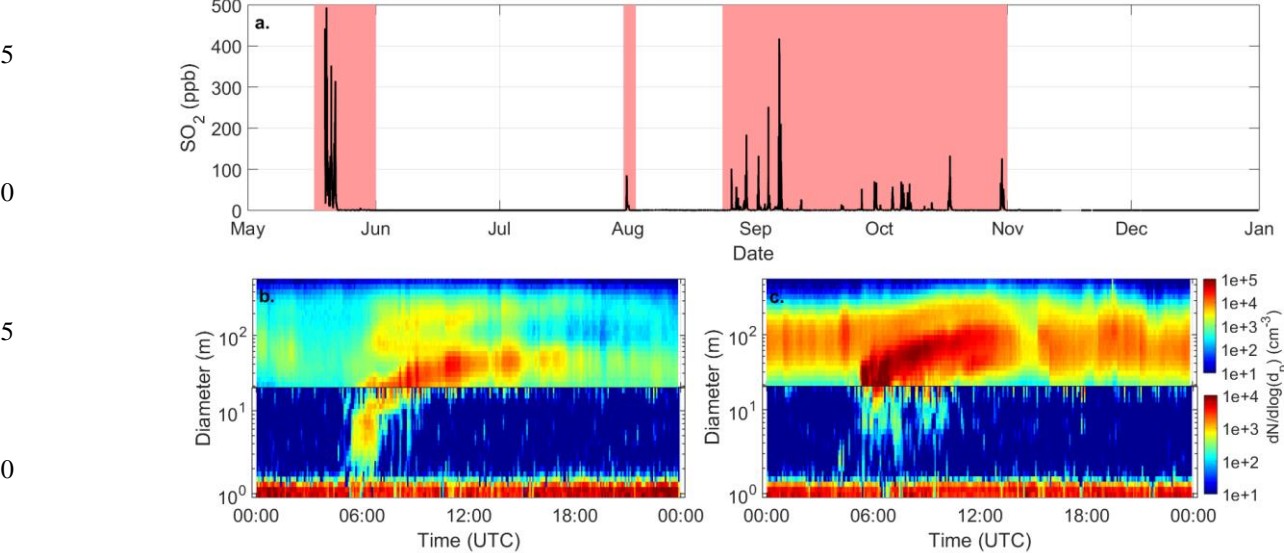

**Figure 1** a. Timeseries of the SO$_2$ mixing ratio measured between May and December 2015 at Maïdo. The shaded areas highlight the three eruptive periods of the Piton de la Fournaise observed during this period. b. Negative ion (lower part) and particle (upper part) number size distributions showing an NPF event occurring on a regular plume day, on May 29th. c. Same as b. for the event detected on May 21st, in strong plume conditions. For b. and c., ion data was derived from AIS measurements, while particle concentrations were measured with the DMPS. Note the different colour scales for ion and particle measurements.

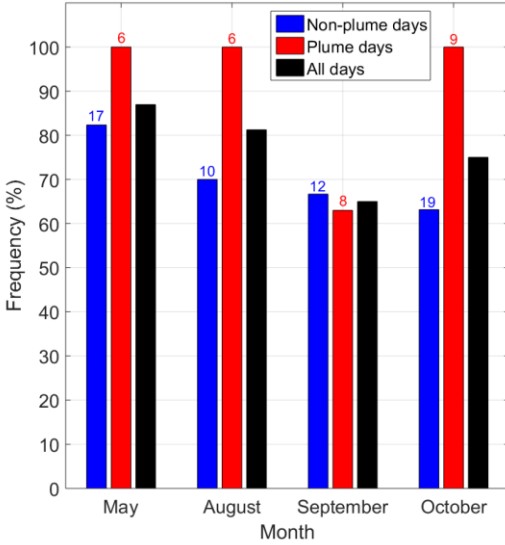

**Figure 2** Frequency of occurrence of NPF at Maïdo. Statistics are shown separately for plume and non-plume days, and total frequencies are also reported. Numbers on the plot indicate, for plume and non-plume conditions, the total number of days included in the statistics.

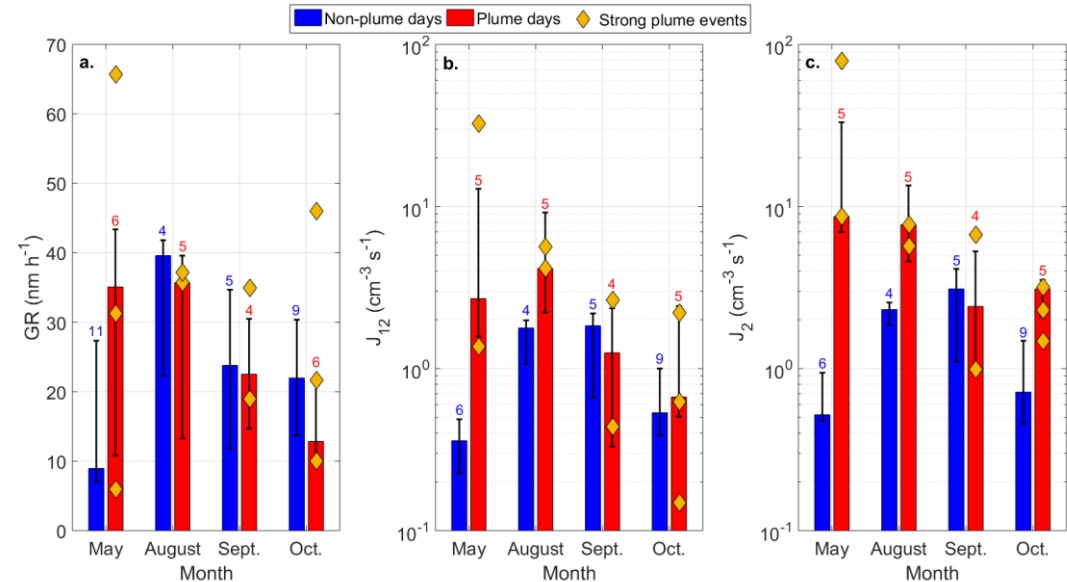

**Figure 3** Monthly medians and percentiles of the NPF event characteristics observed on plume and non-plume days at Maïdo. a. Particle growth rate between 12 and 19 nm ($GR_{12-19}$). Formation rate of b. 12 ($J_{12}$) and c. 2 nm ($J_2$) particles. The bars represent the median of the data, and the lower and upper edges of the error bars indicate 25th and 75th percentiles, respectively. Data collected on strong plume days are included in the statistics reported for plume days and are also highlighted separately. Numbers on each plot indicate, for plume and non-plume conditions, the total number of NPF events included in the statistics.

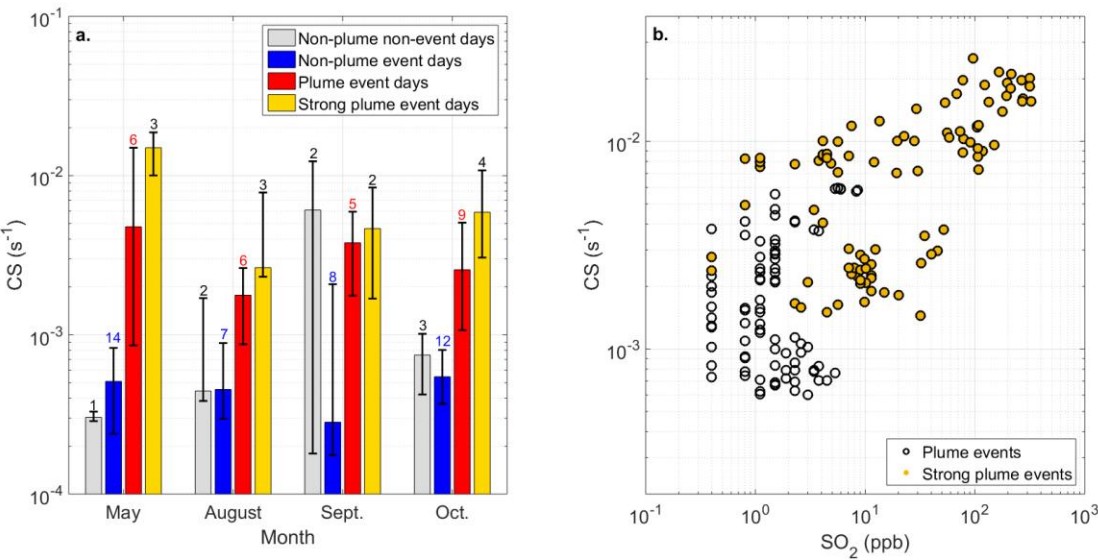

**Figure 4** a. Monthly medians and percentiles of the CS measured prior to NPF hours (i.e. 05:00-07:00 LT). Note that strong plume days are included in the statistics reported for plume days and are also highlighted separately. See Fig. 3 for an explanation of symbols. Numbers on the plot indicate, for plume and non-plume conditions, event and non-event days, the total number of days included in the statistics. b. Link between the CS measured prior to NPF hours and SO$_2$ mixing ratios on plume days. Strong plume days are highlighted with specific markers.

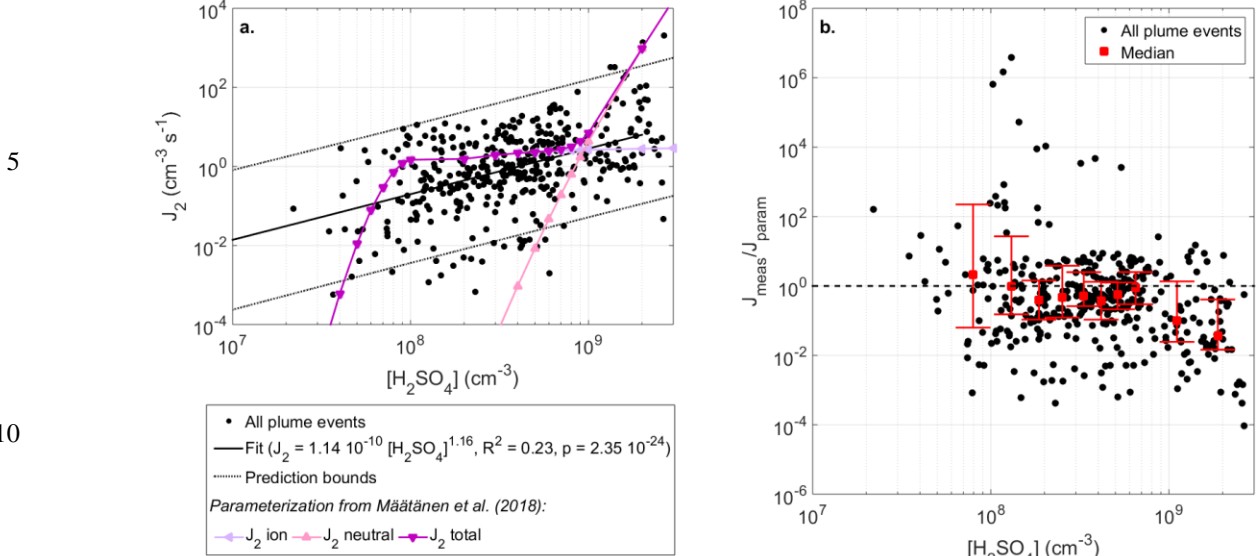

**Figure 5** a. Correlation between the formation rate of 2 nm particles (**$J_2$**) and [$H_2SO_4$] on plume days. A power fit was performed on all available datapoints. The formation rates calculated from the parametrization by Määtänen et al. (2018), which describes the binary nucleation of $H_2SO_4$ – $H_2O$, are also shown, separately for charged ($J_2$ ion) and neutral ($J_2$ neutral) 2 nm clusters. The total formation rate ($J_2$ total) was additionally calculated as the sum of $J_2$ ion and $J_2$ neutral. b. Ratio between the 2 nm particle formation rates derived from DMPS measurements ($J_{meas}$) and the total formation rates derived from the parameterization ($J_{param}$) as a function of [$H_2SO_4$]. Data was also binned with respect to [$H_2SO_4$] (10 bins with equal number of points), and the medians (squares) as well as the 25th/75th percentiles (error bars) of the ratio in each bin are presented.

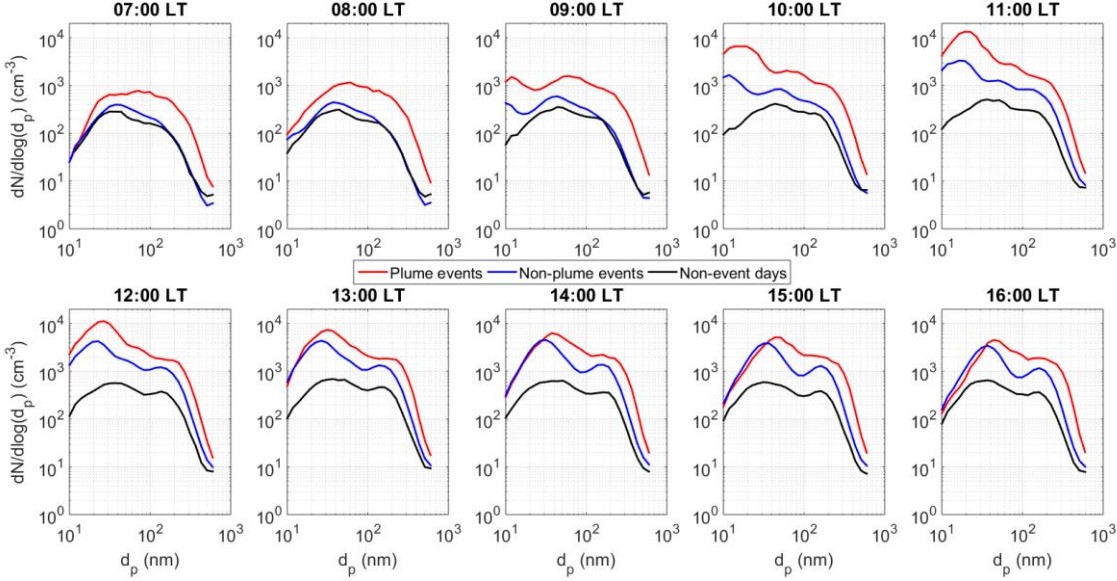

**Figure 6** Hourly medians of the particle size distribution derived from DMPS measurements conducted in the different conditions (non-plume NPF event and non-event days, plume NPF event days) between 07:00 and 16:00 LT.

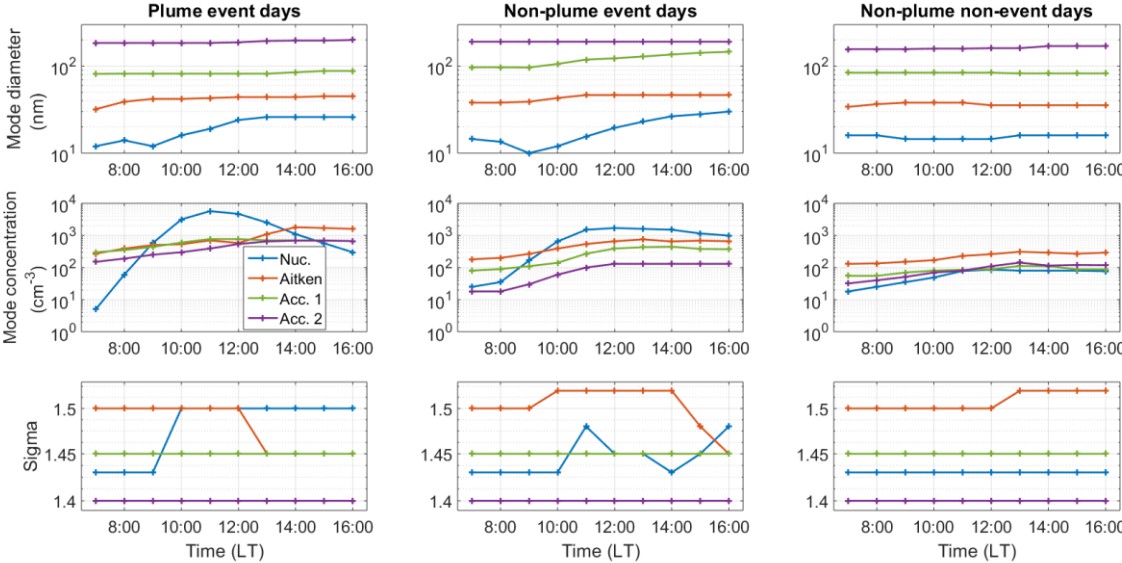

**Figure 7** Variations of the parameters of the Gaussians used to fit the hourly median DMPS size distributions shown on Fig. 6. All the displayed values are also reported in Table A1 of the Appendix.

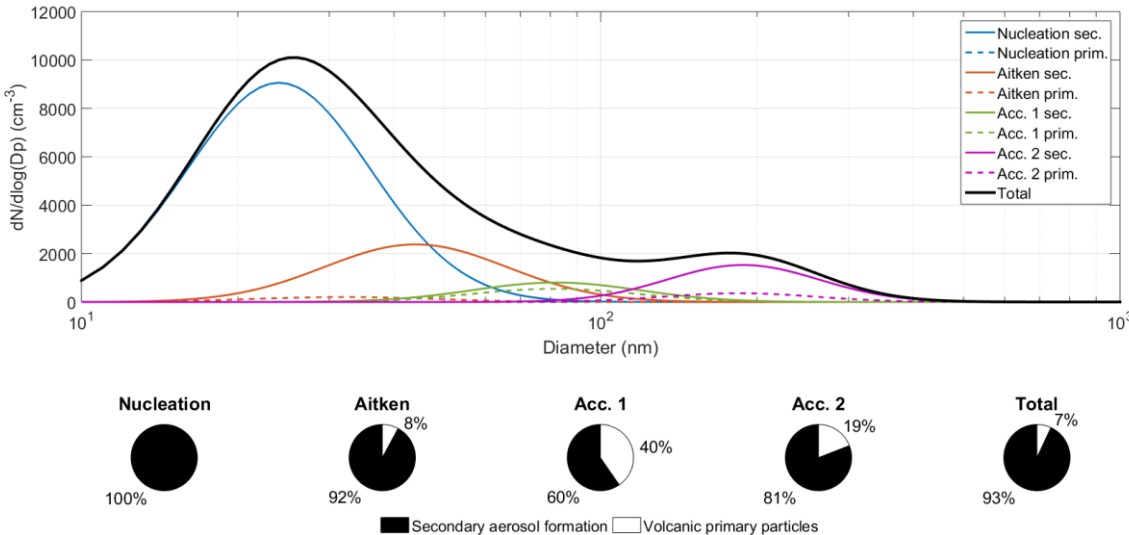

**Figure 8** Size distribution of the particles of volcanic origin reconstructed using the spectra measured on plume and non-plume event days. The contributions of primary (i.e. formed/transformed *via* heterogeneous processes in the very close proximity of the vent) and secondary aerosols are shown separately for each mode on the spectrum and are further highlighted on the pie charts. The contribution of primary aerosols was evaluated based on the spectra measured at 07:00 LT in and off-plume conditions, while the contribution of secondary aerosols was deduced from the maximum concentrations measured for each mode in and off-plume conditions, between 11:00 and 14:00 LT.

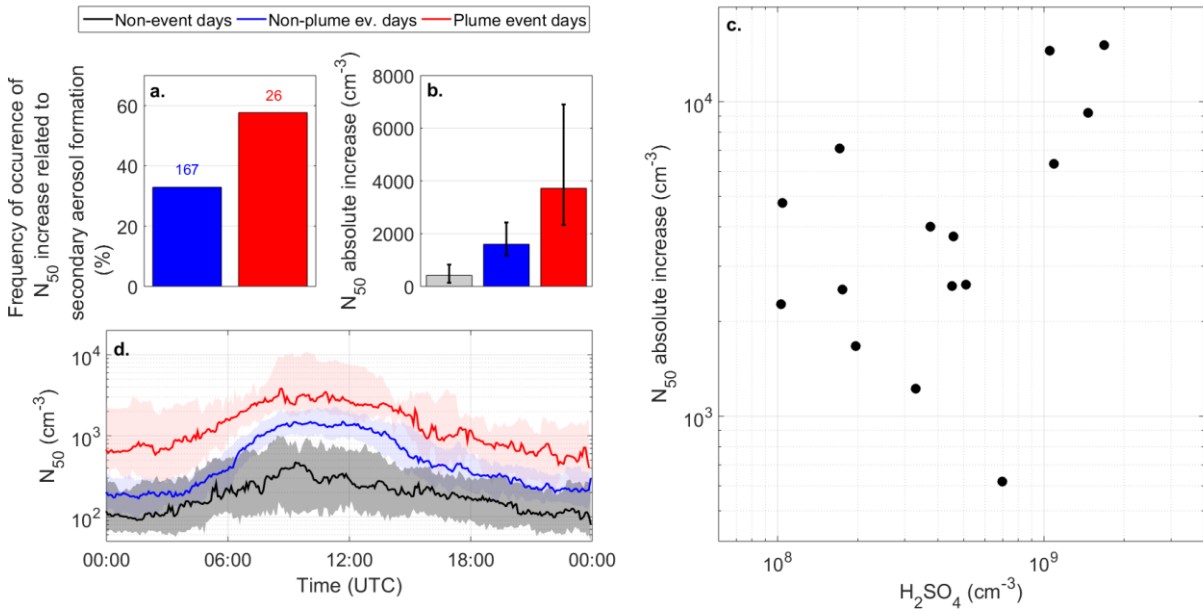

**Figure 9** Increase of potential $> 50$ nm CCN particle concentration ($N_{50}$) during NPF in and off-plume conditions. a. Frequency of occurrence and b. magnitude of the absolute increase. See Fig. 3 for an explanation of symbols. In a., numbers on the plot indicate, for non-plume and plume conditions, the total number of NPF events included in the statistics. c. Link between the increase of $N_{50}$ and [$H_2SO_4$] on plume days. d. Median diurnal variation of $N_{50}$ in the different conditions, including non-plume non-event days. Lower and upper limits of the shaded areas indicate the $25^{th}$ and $75^{th}$ percentiles of the data, respectively.

## Appendix A

**Table A1** Parameters of the Gaussians used to fit the hourly median DMPS size distributions shown on Fig. 6.

| Type of days | Time (LT) | Nucleation mode | | | Aitken mode | | | $1^{st}$ accumulation mode | | | $2^{nd}$ accumulation mode | | |
|---|---|---|---|---|---|---|---|---|---|---|---|---|---|
| | | N (cm$^{-3}$) | Sigma | $d_p$ (nm) | N (cm$^{-3}$) | Sigma | $d_p$ (nm) | N (cm$^{-3}$) | Sigma | $d_p$ (nm) | N (cm$^{-3}$) | Sigma | $d_p$ (nm) |
| Plume event days | 07 :00 | 5 | 1.43 | 12.0 | 270 | 1.50 | 32.0 | 300 | 1.45 | 81.5 | 150 | 1.40 | 185.0 |
| | 08 :00 | 60 | 1.43 | 14.0 | 390 | 1.50 | 39.0 | 350 | 1.45 | 82.0 | 190 | 1.40 | 185.0 |
| | 09 :00 | 580 | 1.43 | 12.0 | 500 | 1.50 | 42.0 | 440 | 1.45 | 82.0 | 250 | 1.40 | 185.0 |
| | 10 :00 | 3150 | 1.50 | 16.0 | 530 | 1.50 | 42.0 | 600 | 1.45 | 82.0 | 300 | 1.40 | 185.0 |
| | 11 :00 | 5700 | 1.50 | 19.0 | 700 | 1.50 | 43.0 | 770 | 1.45 | 82.0 | 390 | 1.40 | 185.0 |
| | 12 :00 | 4700 | 1.50 | 24.0 | 580 | 1.50 | 44.0 | 770 | 1.45 | 82.0 | 540 | 1.40 | 188.0 |
| | 13 :00 | 2500 | 1.50 | 26.0 | 1100 | 1.45 | 44.0 | 700 | 1.45 | 82.0 | 650 | 1.40 | 195.0 |
| | 14 :00 | 1100 | 1.50 | 26.0 | 1800 | 1.45 | 44.0 | 700 | 1.45 | 85.0 | 690 | 1.40 | 197.0 |
| | 15 :00 | 570 | 1.50 | 26.0 | 1700 | 1.45 | 45.0 | 700 | 1.45 | 88.0 | 690 | 1.40 | 197.0 |
| | 16 :00 | 300 | 1.50 | 26.0 | 1600 | 1.45 | 45.0 | 670 | 1.45 | 88.0 | 670 | 1.40 | 200.0 |
| Non-plume event days | 07 :00 | 25 | 1.43 | 14.5 | 180 | 1.50 | 38.0 | 80 | 1.45 | 97.0 | 18 | 1.40 | 191.0 |
| | 08 :00 | 35 | 1.43 | 13.5 | 200 | 1.50 | 38.0 | 90 | 1.45 | 97.0 | 18 | 1.40 | 191.0 |
| | 09 :00 | 165 | 1.43 | 10.0 | 265 | 1.50 | 39.0 | 110 | 1.45 | 96.5 | 30 | 1.40 | 191.0 |
| | 10 :00 | 650 | 1.43 | 12.0 | 390 | 1.52 | 43.0 | 140 | 1.45 | 106.0 | 60 | 1.40 | 191.0 |
| | 11 :00 | 1500 | 1.48 | 15.5 | 540 | 1.52 | 46.5 | 270 | 1.45 | 119.0 | 100 | 1.40 | 191.0 |
| | 12 :00 | 1700 | 1.45 | 19.5 | 660 | 1.52 | 46.5 | 390 | 1.45 | 123.0 | 130 | 1.40 | 191.0 |
| | 13 :00 | 1600 | 1.45 | 23.0 | 750 | 1.52 | 46.5 | 430 | 1.45 | 129.0 | 130 | 1.40 | 191.0 |
| | 14 :00 | 1500 | 1.43 | 26.5 | 650 | 1.52 | 46.5 | 445 | 1.45 | 136.0 | 130 | 1.40 | 191.0 |
| | 15 :00 | 1150 | 1.45 | 28.0 | 690 | 1.48 | 46.5 | 380 | 1.45 | 142.5 | 130 | 1.40 | 191.0 |
| | 16 :00 | 990 | 1.48 | 30.0 | 660 | 1.45 | 46.5 | 370 | 1.45 | 146.0 | 130 | 1.40 | 191.0 |
| Non-plume non-event days | 07 :00 | 18 | 1.43 | 16.0 | 130 | 1.50 | 34.0 | 55 | 1.45 | 84.0 | 32 | 1.40 | 157.0 |
| | 08 :00 | 25 | 1.43 | 16.0 | 135 | 1.50 | 36.5 | 55 | 1.45 | 84.0 | 40 | 1.40 | 157.0 |
| | 09 :00 | 35 | 1.43 | 14.5 | 150 | 1.50 | 38.0 | 70 | 1.45 | 84.0 | 51 | 1.40 | 157.0 |
| | 10 :00 | 49 | 1.43 | 14.5 | 170 | 1.50 | 38.0 | 80 | 1.45 | 84.0 | 70 | 1.40 | 159.0 |
| | 11 :00 | 78 | 1.43 | 14.5 | 230 | 1.50 | 38.0 | 85 | 1.45 | 84.0 | 80 | 1.40 | 159.0 |
| | 12 :00 | 86 | 1.43 | 14.5 | 260 | 1.50 | 35.5 | 89 | 1.45 | 84.0 | 110 | 1.40 | 161.0 |
| | 13 :00 | 80 | 1.43 | 16.0 | 310 | 1.52 | 35.5 | 112 | 1.45 | 82.5 | 142 | 1.40 | 161.0 |
| | 14 :00 | 80 | 1.43 | 16.0 | 290 | 1.52 | 35.5 | 110 | 1.45 | 82.5 | 115 | 1.40 | 170.0 |
| | 15 :00 | 80 | 1.43 | 16.0 | 270 | 1.52 | 35.5 | 88 | 1.45 | 82.5 | 120 | 1.40 | 170.0 |
| | 16 :00 | 77 | 1.43 | 16.0 | 290 | 1.52 | 35.5 | 88 | 1.45 | 82.5 | 118 | 1.40 | 170.0 |

**Author contribution**

K.S. and P.T. organized the measurements, J.-M.M. conducted the measurements, D.P. contributed to the design of the instrumental setup, C.R., B.F., K.S. and A.C. analysed the data, C.R. and K.S. wrote the manuscript, all authors commented on the manuscript.

**Acknowledgements**

This work was performed within the framework of H2020 ACTRIS2 (grant agreement No 654109) and ACTRIS-Fr, and has received financial support from the French programme SNO-CLAP. It has also been funded by the OMNCG/OSU-R program from La Réunion University and by the ANR (ANR STRAP, N°ANR-14-CE03-0004-04). We also want to thank ATMO-

Réunion for providing $SO_2$ data.

**Competing interest**

No competing interest.

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
