# Peer review of "New particle formation in the volcanic eruption plume of the Piton de la Fournaise: specific features from a long-term dataset"

_Atmospheric Chemistry and Physics, 2019_

## Referee Comment (RC1) · Anonymous Referee #1 · 26 Mar 2019

Processes of formation and growth of secondary aerosols, as well as their contribution to the total number of particles (compared to the contribution of primary aerosols) and to the formation of cloud condensation nuclei, are relatively poorly-known in a tropospheric volcanic plume. This study takes advantage of a year-long dataset collected at the high-altitude atmospheric observatory of Maido (La Réunion Island), which was impacted by three eruptions of Piton de la Fournaise volcano in 2015, to investigate such important processes using a statistical approach.

This rich dataset includes ground-based in-situ observations from three complementary instruments: (1) UV fluorescence analysers providing SO2 mixing ratios indicative

of the presence of the volcanic plume, (2) from a Differential Mobility Particle Sizer (DMPS) providing information on the occurrence of new particle formation (NPF) and particle growth rates in the range 10-600 nm, and (3) from an Air Ion Spectrometer (AIS) informing on early stages of NPF.

The paper is well written, figures are clear and interesting data are presented. Given the year-long dataset, the authors can properly determine the occurrence of new particle formation and the typical growth of particles over periods of time which are not impacted by volcanic eruptions. Therefore, they can better identify the impact of volcanic eruptions compared to background conditions. They show the impact of volcanic eruptions on the occurrence of NPF, the importance of photochemistry and the respective contributions of volcanic primary and secondary aerosols on the size distribution of particles. The volcanic impact on particle growth rate seems less obvious however. Results are of strong interest for the atmospheric science community. Nevertheless, as developed in the following, a more robust methodology would be required in some areas to strenghten the approach and strongly support the results:

- Classification of plume- vs non-plume days :

As the classification is critical for the statistical study, more details and illustrations are missing to describe and validate the classification of plume- vs non-plume days. As SO2 represents indeed a clear tracer of the volcanic plume, a time series of SO2 mixing ratio values at Maido with highlighted volcanic events would be welcome in order to evaluate the amplitude of background variations in SO2 mixing ratios.

As this is the root of the paper, an illustration with AIS and DMPS observations for one representative strong plume day and one weakly influenced plume day before statistical representation of Fig. 1 would be required.

- Selection of plume-days, page 7, lines 19-27 :

If I understand correctly, selected days are considered as 'plume-days' when at least

one of the hourly averages of the SO2 mixing ratio exceeds 1 ppb over the 5 hours of interest each day (between 6 :00 and 11 :00 LT). The volcanic plume was detected during the 5 hours of the time window of interest for only 20 of the 36 'plume-days'. I am wondering if the authors should not restrict their study to these 'fully volcanically-influenced days' ? If not, they should assess the impact of mixing in their study 'plume-days' hours without any volcanic plume. This choice may artificially tend to decrease the difference between plume- and non plume-days.

- Start time of NPF events, page 9 :

Why are the detection and evaluation of the start time made by a visual inspection ? What is the difficulty in automating the detection of a concentration increase in the 1.5-2.5 nm range ? Visual inspection is subject to large uncertainty and raise questions on the accuracy and reproducibility of the obtained results. An illustrative example would be also welcome to see how strong are the AIS/DMPS signals for days only poorly-contaminated by volcanic plumes.

- Particle growth rate, pages 10-11 :

Page 10, lines 3-8 : The authors do not highlight any impact of the volcanic plume on the particle growth rate between 12 and 19 nm. The interpretation of the authors is that it may be difficult to clearly identify the impact of volcanic plumes at the Maido Observatory as the atmospheric dynamics is complex around this site and there may be an importation of growing particles likely transported to this site. The same processes (of imported particles, including potentially biomass burning aerosols as mentionned for CS variations in Sept and Oct) could also bias the observations of J2 and J12 ? The authors should comment on this and propose some solutions to 'clean' data by removing periods with strong influences by other sources of aerosols (urban, biomass burning, etc..).

Clearly higher values of J2 and J12 values are not observed under volcanic influence for the month of Sept. A clear volcanic signature is not identified either for J2 values

for the month of Oct (with also surprisingly very spread J2 values for strong plume days). The authors should describe these discrepancies in the text and provide some interpretations or suggestions of interpretation (impact of biomass burning activities, or others ?).

By contrast, page 10, lines 9-14 : why is observed in May so much increase in J2 and J12 values for plume-days compared to non-plume days ? Is there a specificity of the volcanic events, or of the meteorological conditions occuring in May? Opposite case: why is not observed an obvious distinct behaviour of plume-days in Oct ?

More generally, the authors should discuss the advantages and also the disadvantages or limitations to have data collected at a high altitude atmospheric observatory and the potential biases that may affect the results at such a site (including complex atmospheric dynamics, fluctuating relative humidity, is it easier or not to identify imported species, a less polluted background or not, etc. . .).

- Which is the impact of relative humidity on NPF? As relative humidity is measured at Maido, is there any correlation with NPF? Are observed higher RH values during plume-days (as the Piton de la Fournaise plume may be rich in volcanic water vapour) or not ?

- Impact of condensation sink (CS) page 11 :

Right of Fig. 3 : in the plot of CS vs SO2 mixing ratio for plume-days and strong-plume days, could it be added non-plume days to assess if obvious differences are observed between plume- and non plume-days in this representation ?

Left of Fig. 3 : how do the authors explain the large CS observed in Sept and Oct for non-plume days ?

- Relationship between J2 and [H2SO4], page 12 :

According to Fig. 4, a correlation relationship between J2 and [H2SO4] is not obvious : data points are very scattered, as illustrated by the very low value of $R^2$ of 0.21 and

0.11 for all plume and strong-plume days respectively. In this context, is it meaningful to try to fit anyway a correlation relationship and estimate k and a coefficients?

Moreover, except higher concentrations of H2SO4, data associated to strong-plume days do not seem to present a very different relationship between J2 and [H2SO4] (Fig. 4a). The weak difference in the relationship which is retrieved seems just to result from the influence of 3 points, potentially outliers ? If these points would have been represented in black, and not in yellow, it would be very difficult to distinguish any different behaviour.

Minor comments :

- 'Active volcanic plume' : I do not understand this term. Given lines 31-32 in the introduction, I am wondering if the authors may want to refer to a volcanic plume emitted during an eruption compared to passive degassing emitted out of eruptive periods. If so, please refer rather to 'volcanic eruption plume'

- Page 3, lines 5-8 : 'primary particles are fragment of ash while secondary particles. . .' : Volcanic primary particles do not include only ash particles but also sulfate aerosols, as illustrated by near-source measurements (e.g. refer to first publications on this matter which include Allen et al., 2002 ; Mather et al., 2003, 2004, etc..).

- 'Here we report observations of NPF performed at the high-altitude observatory of Maïdo (2165 m a.s.l., La Réunion Island) between 1st January and 31st December 2015. During this period of time, 3 effusive eruptions of the Piton de la Fournaise, located 39 km away from the station, were observed and documented, resulting in 36 days of measurement in volcanic plume conditions to be compared with 250 "non -plume days'. 250 + 26 = 276 days, what happens with the missing 89 (=365-276) days ?

- There are many references to a study in preparation (Sahyoun et al., in prep) which is presented as an earlier work: has this paper been submitted to a journal with open

discussion where it would be accessible or has it been published since then? If yes, please update so that the reader can have access to this manuscript.

Text :

- Abstract is very long, if possible you should try to shorten it (possibly remove the mention to the correlation relationship between J2 and H2SO4 concentration which does not seem obvious (as developed above)

- Please reformulate these sentences for clarity :

- abstract, Page 1, line 17 : 'as those form the baseline to calculate..'

- abstract, Page 1, line 30 : 'recorded in the different conditions': recorded in the different conditions described thereafter..

- abstract, Page 1, line 26-27 : 'compared to non-plume days, during which condensable species were in contrast transported from lower altitude by the mean of convective processes' : it is difficult to understand the meaning of this sentence if we have not read the manuscript yet

- Page 2, lines 21-22 : 'the radiative forcing. . . still has a large uncertainty'

- Page 11, line 16 : 'loss rate of the vapours' ? What do you mean by 'vapours' ?

---

## Referee Comment (RC2) · Anonymous Referee #2 · 24 Apr 2019

This paper deals with an important, yet very little-studied, topic: new particle formation (NPF) in a volcanic plume. Therefore, the paper can be considered highly relevant and also original. The conducted analysis is mostly scientifically sound, and text is relatively well written. I have a few comments that should be addressed before I can recommend the acceptance of this paper for publication.

Main scientific issues

Section 2.2. The authors should discuss briefly the uncertainties and limitations of the equations 1 to 4 in calculating the particle formation ($J$) and growth ($GR$) rates in their data. First, these equations have been developed originally for regional NPF, in which

formation and growth of particles is assumed to take place relatively homogeneously over large spatial scales. This is apparently not the case in plumes where, among other things, various transport effects on J and GR should be taken into account. Second, experimental limitations cause further uncertainties in determining J and GR. For example, using coagulation sink at 12 nm for all particles in the size range 12-19 nm in equation 1 causes some overestimation of coagulation losses, which results in underestimating J12. Also, Calculating J2 from J12 would require knowing GR in the size range 2-12 nm rather than that in the size range 12-19 nm. While it is impossible to take into account the above issues to correct the data, the authors should at the very least discuss these issues briefly in section 2.2. If possible, the authors could also estimate whether resulting uncertainties are important or not with respect to their results.

Section 3.2.2. In this work, neither J2 nor H2SO4 concentration were measured directly, but were derived from other measured quantities, resulting in potentially large uncertainties in their values. This has implications which are not mentioned in the paper. First, how reliable is the observed relation between J2 and H2SO4 concentration, and how meaningful is it to compare this relation with those observed in studies were J and H2SO4 concentration were measured directly? Second, how meaningful is it compare J obtained here with parameterized J due to binary water-sulfuric acid nucleation as a function of H2SO4 concentration? Does this comparison tell anything about nucleation mechanism?

There are a few issues related to the particle growth that need some clarifications. First, did the authors consider particle growth from one mode to another when estimating the relative contributions of primary and secondary particles in each mode? This remains a bit unclear when reading the results. Second, the authors do not tell what were the typical air mass transport times from the volcano to the measurement site. This is important because for the reported particle growth rates (Fig. 2a), it takes a while before particles formed in the plume are able to growth into the Aitken mode, and for several hours before they can reach the minimum CCN size (assumed >50 nm here)

or the accumulation mode. Is it feasible that particle formed by NPF in the volcanic plume reach these sizes by the time measurements were conducted? Third, while I agree with the authors that volcanic emissions are able to boost particle growth by e.g. heterogenous reactions of SO2 on particle surfaces, there seems to be some inconsistences in the storyline: on one hand the authors state that the plume appear not to influence the particle growth (section 3.1.3), and on the other hand they state that particle growth in the plume increased both modal (section 3.3.1) and CCN (section 3.3.2) concentrations.

Minor/technical issues

Page 7, line 4: "...when global radiation >50 ...". Something is missing from here (was?).

The format of providing the time difference (i.e. 2h10) in section 3.1.2 seems strange to me. Is this a correct way of expressing the time difference?

Page 10, line 2: "GR12-19 showed an important variability, ...". What do the authors mean by "important" here?

Excluding the last paragraph of section 4, the text in that section mainly summarizes the results discussed eaerlier in the paper. As a results, an appropriate title of this section would be "4. Summary and Conclusions".

Would it be possible to change the lines and marks with yellow color in Figures into some other, more easily visible color?

---

## Author Comment (AC1) · 4 Jul 2019

We thank Referee #1 for his comments and suggestions that, we believe, have helped improving the manuscript. We have addressed the comments point by point below. In addition, the errorbars shown on Fig. 9.b were modified, as those did not correspond to the actual variability of $N_{50}$ absolute increase in the original version of the manuscript (former Fig. 8.b).

**Comment 1**: Classification of plume- vs non-plume days:
As the classification is critical for the statistical study, more details and illustrations are missing to describe and validate the classification of plume- vs non-plume days. As $SO_2$ represents indeed a clear tracer of the volcanic plume, a time series of SO2 mixing ratio values at Maido with highlighted volcanic events would be welcome in order to evaluate the amplitude of background variations in SO2 mixing ratios.
As this is the root of the paper, an illustration with AIS and DMPS observations for one representative strong plume day and one weakly influenced plume day before statistical representation of Fig. 1 would be required.

**Reply 1**: Adding a figure to support the method we followed to detect the volcanic eruption plume at the station, and to illustrate as well NPF events occurring in such unusual conditions, is in fact a good suggestion. Hence, we have included a new figure (Fig. 1) in the revised version of the manuscript, which displays a. The timeseries of the SO2 mixing ratio with some indication of the eruptive periods, b. AIS and DMPS observations showing an NPF event on a regular plume day and c. on a strong plume day.

**Comment 2**: Selection of plume-days, page 7, lines 19-27 : If I understand correctly, selected days are considered as 'plume-days' when at least one of the hourly averages of the SO2 mixing ratio exceeds 1 ppb over the 5 hours of interest each day (between 6 :00 and 11 :00 LT). The volcanic plume was detected during the 5 hours of the time window of interest for only 20 of the 36 'plume-days'. I am wondering if the authors should not restrict their study to these 'fully volcanically influenced days' ? If not, they should assess the impact of mixing in their study 'plume days' hours without any volcanic plume. This choice may artificially tend to decrease the difference between plume- and non plume-days.

**Reply 2**: We agree with the fact that including days with very short plume occurrence in the statistics did not give the most accurate picture of the volcanic eruption plume signature on NPF and related variables of interest. Hence, we have revised our classification to include only the days when the volcanic plume was detected over at least 3 hours between 06:00 and 11:00 LT. These days represent 80% of the former plume days (29/36); remaining 7 days with short plume occurrence, which are now excluded from the revised classification, were found in September and October. Corresponding statistics and results have been updated throughout the manuscript, but it is worth noticing that the main conclusions of this work were not affected.
Note that plume days were not restricted to the days when the plume was observed over the five hours of interest to allow a fair compromise between the number of plume days, that we wanted to keep sufficient for the relevance of our analysis, and significant influence of the volcanic eruption plume. Effect of the more "intense" plume conditions, both in terms of $SO_2$ levels and time duration of the plume occurrence, is further investigated by the mean of the so-called strong plume days.

**Comment 3**: Start time of NPF events, page 9:
Why are the detection and evaluation of the start time made by a visual inspection?
What is the difficulty in automating the detection of a concentration increase in the 1.5-2.5 nm range?
Visual inspection is subject to large uncertainty and raise questions on the accuracy and reproducibility of the obtained results. An illustrative example would be also welcome to see how strong are the AIS/DMPS signals for days only poorly-contaminated by volcanic plumes.

**Reply 3**: The detection of the concentration increase could undoubtedly be automated, and recent studies have by the way reported different methods to allow for an automatic monitoring and description of NPF (eg: Hussein et al., 2005; Dall'Osto et al., 2017; Dada et al., 2018). Nonetheless, to our knowledge, the visual approach for the identification and analysis of NPF has, to date, been the most commonly

used and is still popular. This is for instance illustrated by the very recent study by Hakala et al. (2019), who visually determined 4 different times to describe the progression of each NPF event. There is for sure an uncertainty which is associated to this analysis "by eye", but we believe that such approach is less risky than automated ones with respect to more "critical" errors. In our case, such error could for instance be the detection of a concentration increase which is not connected to any clear NPF event.

As previously mentioned in Reply 1, an additional figure was added (Fig. 1) in the revised version of the manuscript to illustrate the NPF process in plume conditions.

**Comment 4**: Particle growth rate, pages 10-11:

**Comment 4.A**: Page 10, lines 3-8: The authors do not highlight any impact of the volcanic plume on the particle growth rate between 12 and 19 nm. The interpretation of the authors is that it may be difficult to clearly identify the impact of volcanic plumes at the Maido Observatory as the atmospheric dynamics is complex around this site and there may be an importation of growing particles likely transported to this site. The same processes (of imported particles, including potentially biomass burning aerosols as mentioned for CS variations in Sept and Oct) could also bias the observations of J2 and J12? The authors should comment on this and propose some solutions to 'clean' data by removing periods with strong influences by other sources of aerosols (urban, biomass burning, etc..).

**Reply 4.A**: It is absolutely true that, while affecting the determination of the particle growth, complex topography and atmospheric dynamics in turn also affect the calculation of particle formation rates. This is now clearly stated in Section 3.1.3: *"However, assessing the real effect of these specific conditions on the particle formation and growth is challenging"*. Also, following a comment from Referee #2, we have included an additional discussion in Section 3.1.3 to explain the limited effect of the plume on measured growth rates: *"This observation is most likely related to the fact that not only the amount of precursor vapours (including for instance SO2, see Fig. 1.a) was increased in the volcanic plume, but also the number concentration of the particles to grow, from both primary and secondary origin, as also reflected in the variations of the CS (Section 3.2.1) and discussed later in Section 3.3.1"*.

Instead of cleaning the data and removing some periods, the best would be to be able to evaluate the contribution of the different sources to the observed NPF events, and in turn to GR and J, in the same manner we have discussed the particle size distributions in Section 3.3.1. This is however much more complex for J and GR, and such detailed analysis of the impact of different conditions (anthropogenic air masses, biomass burning…) is anyway beyond the scope of the present study. Instead, our main goal was only to evaluate the effect of the plume conditions on J and GR in comparison with non-plume days. For that purpose, we have used a statistical approach, which allows for the comparison of a number of plume and non-plume days which is assumed to be sufficient to smooth the specificities of each single event with respect to all conditions other than plume occurrence.

**Comment 4.B**: Clearly higher values of J2 and J12 values are not observed under volcanic influence for the month of Sept. A clear volcanic signature is not identified either for J2 values for the month of Oct (with also surprisingly very spread J2 values for strong plume days). The authors should describe these discrepancies in the text and provide some interpretations or suggestions of interpretation (impact of biomass burning activities, or others?).

By contrast, page 10, lines 9-14: why is observed in May so much increase in J2 and J12 values for plume-days compared to non-plume days? Is there a specificity of the volcanic events, or of the meteorological conditions occuring in May? Opposite case: why is not observed an obvious distinct behaviour of plume-days in Oct?

**Reply 4.B**: It is true that we should have commented more on the comparison between the formation rates measured in and off- plume conditions, as different trends are effectively highlighted in this study. We have thus included a paragraph in Section 3.2.1, which discusses the variability of CS, together with that of $SO_2$ mixing ratio, as we believe they are key parameters driving the variations of $J_{12}$ and $J_2$. This discussion also refers to the variations of the particle growth rate, in response to one of the comments from Referee #2: *"In addition, the balance between the amount of $SO_2$ and the magnitude of the CS most likely influenced the strength of the observed events, and explained in specific the variable trends*

*highlighted earlier in the comparison of the particle formation rates calculated on plume and non-plume days (see Sect. 3.1.3, Fig. 3.b and c). Indeed, as reported previously, the largest CS increase observed between non-plume and plume NPF event days occurred in May, when SO$_2$ mixing ratios were also the highest (Fig. 1), with a median of 26.7 ppb [25$^{th}$ percentile: 1.1 ppb; 75$^{th}$ percentile: 120.5 ppb] calculated during nucleation hours (06:00 and 11:00 LT). We may thus hypothesize that the resulting conditions were highly favourable to NPF, and not only lead to high NPF frequency (Fig. 2), but also to stronger events, with increased particle formation rates compared to non-plume days (Fig. 3.b and c). In September and October, the median CS measured in plume conditions were comparable to that observed in May (Fig. 4.a), but SO$_2$ mixing ratios were in contrast lower during nucleation hours, with medians around 3.4 ppb [1.5 ppb; 5.6 ppb] and 3.8 ppb [1.9 ppb; 16.9 ppb], respectively. This most likely resulted in less favourable conditions for NPF than in May, which in turn did not enhance the particle formation rates compared to non-plume days. Higher CS observed on plume days also supported the fact that in plume conditions, as suggested in the previous section, the number of particles to grow was increased compared to non-plume days, and the concurrent strengthening of the precursor source rate was on average not sufficient to result in faster particle growth. Nonetheless, while it was possible to evidence the abovementioned trends with our statistical approach, one should keep in mind that both the occurrence and characteristics of NPF are likely to be affected at very short time scales due to the variable nature of the volcanic eruption. Deeper investigation of the effect of the volcanic eruption plume on NPF would thus require more detailed analysis of the event to event variability, which was however beyond the scope of the present work".*

**Comment 4.C**: More generally, the authors should discuss the advantages and also the disadvantages or limitations to have data collected at a high altitude atmospheric observatory and the potential biases that may affect the results at such a site (including complex atmospheric dynamics, fluctuating relative humidity, is it easier or not to identify imported species, a less polluted background or not, etc…).

**Reply 4.C**: We believe that the main specificities of a site such as the Maïdo which were relevant to the present work have been mentioned in the manuscript, including in particular the complex topography and atmospheric dynamics which affect the transport of both NPF gaseous precursors an growing/pre-existing particles, with consequent effect on NPF characteristics, particle size distribution and in turn CCN population. Deeper analysis of the advantages/disadvantages of high altitude observatories is behind the scope of the present work; it is however of high interest, and is the main focus of a review dedicated to the observation of NPF from such high altitude sites, currently in preparation.

**Comment 5**: Which is the impact of relative humidity on NPF? As relative humidity is measured at Maido, is there any correlation with NPF? Are observed higher RH values during plume-days (as the Piton de la Fournaise plume may be rich in volcanic water vapour) or not?

**Reply 5**: The effect of RH on NPF is not plain to understand, as contrasting observations have been reported, regarding in specific its influence on the particle formation rates (e.g. Birmili et al., 2003; Duplissy et al., 2016). We have investigated the variations of RH observed at Maïdo on event days, in and off-plume conditions, and on non-event days, together with that of temperature and radiation, which are also reported as key meteorological parameters. The results of this analysis are reported at the beginning of Section 3.2: *"NPF has been previously reported to be influenced by various atmospheric parameters, including solar radiation, temperature (Dada et al., 2017), as well as RH, which effect on the process is certainly the less evident to predict and understand (e.g. Birmili et al., 2003; Duplissy et al., 2016). In the frame of the present analysis, the median diurnal variations of the abovementioned parameters reported on Fig. S1 (in the Supplementary) did not highlight any specificity for the events observed on plume days, and displayed similar behaviour in and off-plume conditions".*
A corresponding figure showing the median diurnal patterns of temperature, RH and global radiation measured in the different conditions (non-event days, event days in and off-plume) was included in the Supplementary; note that all the figures originally shown in the different Appendices were moved to the Supplementary in order not to multiply the number of Appendices.

**Comment 6**: Impact of condensation sink (CS) page 11:

Right of Fig. 3: in the plot of CS vs SO2 mixing ratio for plume-days and strong-plume days, could it be added non-plume days to assess if obvious differences are observed between plume- and non plume-days in this representation?
Left of Fig. 3: how do the authors explain the large CS observed in Sept and Oct for non-plume days?

**Reply 6**: It was unfortunately not possible to include non-plume days in Fig. 4.b (former Fig. 3.b) because SO$_2$ mixing ratios were mostly below the detection limit of the instrument outside of the eruptive periods, as indicated in Section 2.1: *"The detection limit of the instrument was about 0.5 ppb, which is above the usual SO2 mixing ratios encountered at Maïdo outside of the eruptive periods of the Piton de la Fournaise (see Fig. A1 in Foucart et al. 2018)"*. This is now also recalled in Section 2.3: *"The relatively low SO$_2$ mixing ratios observed outside of the eruptive periods, mostly below the detection limit of the instrument, reflect the low pollution levels characteristic of this insular station, located at high altitude in a region rarely subject to significant influence of pollution from continental origin"*.
Detailed investigation of the origin of the large CS observed on non-event days in September and October was behind the scope of the present work, nonetheless this observation is already briefly discussed in the manuscript (Section 3.2.1) : *"Indeed, comparable median CS were observed in August regardless the occurrence of NPF later during the day, higher values were in contrast obtained on event days in May, while the opposite was observed in September and October, most likely related to biomass burning activity in South Africa and Madagascar during austral spring (Clain et al., 2009; Duflot et al., 2010; Vigouroux et al., 2012)."*

**Comment 7**: Relationship between J2 and [H2SO4], page 12:
According to Fig. 4, a correlation relationship between J2 and [H2SO4] is not obvious: data points are very scattered, as illustrated by the very low value of R2 of 0.21 and 0.11 for all plume and strong-plume days respectively. In this context, is it meaningful to try to fit anyway a correlation relationship and estimate k and a coefficient?
Moreover, except higher concentrations of H2SO4, data associated to strong-plume days do not seem to present a very different relationship between J2 and [H2SO4] (Fig. 4a). The weak difference in the relationship which is retrieved seems just to result from the influence of 3 points, potentially outliers? If these points would have been represented in black, and not in yellow, it would be very difficult to distinguish any different behaviour.

**Reply 7**: The correlation between J$_2$ and [H$_2$SO$_4$] derived from data measured on all plume days is definitely moderate but still significant, as indicated by the corresponding p-value ($2.35 \times 10^{-24}$). Regarding strong plume days, we found a mistake in the selection of the data points for the fitting procedure. Using the correct data finally lead to very similar fit parameters to that obtained for all plume days, which is now explicitly mentioned in the revised version of the manuscript: *"the relationship between J$_2$ and [H2SO4] did not appear to be significantly different on strong plume days compared to regular plume conditions"*. For that reason (and also for more clarity) strong plume days are not any longer highlighted on Fig. 5. Additional discussion regarding the contributions of charged and neutral nucleation pathways was also included in Section 3.2.2, based on the use of the parameterisation by Määtänen et al. (2018): *"As evidenced on Fig. 5.a, the total formation rate of 2 nm-clusters was mostly explained by ion induced nucleation for [H2SO4] below ~ $8 \times 10^8$ cm-3, while neutral pathways seemed to explain the observations at larger sulfuric acid concentrations"*.
Also, since the results reported in Sections 3.1 and 3.2 only revealed limited signature of strong plume conditions on NPF characteristics (and related parameters of interest), we have decided not to put any focus on these specific days in the last Section (3.3) in order to make our message as clear as possible.

**Minor comments:**

**Comment 1**: 'Active volcanic plume': I do not understand this term. Given lines 31-32 in the introduction, I am wondering if the authors may want to refer to a volcanic plume emitted during an

eruption compared to passive degassing emitted out of eruptive periods. If so, please refer rather to 'volcanic eruption plume'

**Reply 1:** Changed throughout the manuscript.

**Comment 2:** Page3,lines5-8: 'primary particles are fragment of ash while secondary particles...' : Volcanic primary particles do not include only ash particles but also sulfate aerosols, as illustrated by near-source measurements (e.g. refer to first publications on this matter which include Allen et al., 2002 ; Mather et al., 2003, 2004, etc..).

**Reply 2:** We thank the reviewer for pointing out this omission. This has been addressed in the revised version of the manuscript:
- In the introduction: *"Primary sulphate aerosols of volcanic origin were also evidenced by near source measurements conducted at Masaya volcano by Allen et al. (2002), who were however not able to identify their precise mechanisms of formation. Several pathways were later suggested for the formation of $H_2SO_4$ at the vent, including catalytic oxidation of $SO_2$ inside the volcanic dome Zelenski et al. (2015), high temperature chemistry in the gas phase (Roberts et al., 2019), as well as aqueous production of $H_2SO_4$ from $SO_2$ (Tulet et al., 2017). $H_2SO_4$ produced by the mean of the aforementioned reactions is expected to contribute to the formation and growth of particles in the close vicinity of the volcano, which are in turn assimilated to primary volcanic aerosols"*.
- In Section 3.2.1: *"This nomenclature is consistent with earlier results from Allen et al. (2002), who reported the presence of primary sulphate aerosols at Masaya volcano"*.

**Comment 3:** 'Here we report observations of NPF performed at the high-altitude observatory of Maïdo (2165 m a.s.l., La Réunion Island) between 1st January and 31st December 2015. During this period of time, 3 effusive eruptions of the Piton de la Fournaise, located 39 km away from the station, were observed and documented, resulting in 36 days of measurement in volcanic plume conditions to be compared with 250 "non -plume days'. 250 + 26 = 276 days, what happens with the missing 89 (=365-276) days?

**Reply 3:** It is true that the reported numbers were somewhat confusing and needed some clarification, now available in Section 2.3: *"In the end, after filtering the data for instrument malfunctioning and / or absence of measurements (71 days in total), 29 plume days and 250 non-plume days were included in the analysis. The 15 remaining days, with late or short plume occurrence, will not be further discussed"*.

**Comment 4:** There are many references to a study in preparation (Sahyoun et al., in prep) which is presented as an earlier work: has this paper been submitted to a journal with open discussion where it would be accessible or has it been published since then? If yes, please update so that the reader can have access to this manuscript.

**Reply 4:** This paper was indeed not published when we first submitted our manuscript. We have now included the reference:
Sahyoun, M., Freney, E., Brito, J., Duplissy, J., Gouhier, M., Colomb, A., Dupuy, R., Bourianne, T., Nowak, J. B., Yan, C., Petäjä, T., Kulmala, M., Schwarzenboeck, A., Planche, C. and Sellegri, K. : Evidence of new particle formation within Etna and Stromboli volcanic plumes and its parameterization from airborne in-situ measurements, J. Geophys. Res.-Atmos, https://doi.org/10.1029/2018JD028882, 2019.

**Comment 5:** Abstract is very long, if possible you should try to shorten it (possibly remove the mention to the correlation relationship between $J_2$ and $H_2SO_4$ concentration which does not seem obvious (as developed above).

**Reply 5:** The abstract is indeed quite long, but we believe it gives the opportunity to the readers (who are nowadays often busy!!) to quickly get the storyline and main results of this work.

**Comment 6**: Please reformulate these sentences for clarity:
1) abstract, Page 1, line 17 : 'as those form the baseline to calculate..'
2) abstract, Page 1, line 30 : 'recorded in the different conditions': recorded in the different conditions described thereafter..
3) abstract, Page 1, line 26-27: 'compared to non-plume days, during which condensable species were in contrast transported from lower altitude by the mean of convective processes': it is difficult to understand the meaning of this sentence if we have not read the manuscript yet.
4) Page 2, lines 21-22 : 'the radiative forcing... still has a large uncertainty'
5) Page 11, line 16 : 'loss rate of the vapours' ? What do you mean by 'vapours'?

**Reply 6:**
1) This part of the sentence was removed as the information it contains is not of the highest importance for the abstract. This topic is further discussed in the introduction.
2) Changed to: "based on the analysis and fitting of the particle size distributions recorded in and off-plume conditions".
3) For more clarity this part of the sentence was also removed from the abstract; corresponding results are discussed in details in the manuscript.
4) Changed to "the radiative forcing associated to these effects (usually referred to as "indirect effect") is known with a still large uncertainty".
5) Changed to "the strengthened loss rate of the condensing vapours involved in particle formation and growth".

**References:**

Birmili, W., Berresheim, H., Plass-Dülmer, C., Elste, T., Gilge, S., Wiedensohler, A., and Uhrner, U.: The Hohenpeissenberg aerosol formation experiment (HAFEX): a long-term study including size-resolved aerosol, H2SO4, OH, and monoterpenes measurements, Atmos. Chem. Phys., 3, 361-376, https://doi.org/10.5194/acp-3-361-2003, 2003.

Dada, L., Chellapermal, R., Buenrostro Mazon, S., Paasonen, P., Lampilahti, J., Manninen, H. E., Junninen, H., Petäjä, T., Kerminen, V.-M., and Kulmala, M.: Refined classification and characterization of atmospheric new-particle formation events using air ions, Atmos. Chem. Phys., 18, 17883-17893, https://doi.org/10.5194/acp-18-17883-2018, 2018.

Dall'Osto, M., Beddows, D. C. S., Asmi, A., Poulain, L., Hao, L., Freney, E., Allan, J. D., Canagaratna, M., Crippa, M., Bianchi, F., de Leeuw, G., Eriksson, A., Swietlicki, E., Hansson, H. C., Henzing, J. S., Granier, C., Zemankova, K., Laj, P., Onasch, T., Prevot, A., Putaud, J. P., Sellegri, K., Vidal, M., Virtanen, A., Simo, R., Worsnop, D., O'Dowd, C., Kulmala, M. and Harrison, Roy M.: Novel insights on new particle formation derived from a pan-european observing system, Scientific Reports, 8, 1482, 2018.

Duplissy, J., Merikanto, J., Franchin, A., Tsagkogeorgas, G., Kangasluoma, J., Wimmer, D., Vuollekoski, H., Schobesberger, S., Lehtipalo, K., Flagan, R. C., Brus, D., Donahue, N. M., Vehkamäki, H., Almeida, J., Amorim, A., Barmet, P., Bianchi, F., Breitenlechner, M., Dunne, E. M., Guida, R., Henschel, H., Junninen, H., Kirkby, J., Kürten, A., Kupc, A., Määttänen, A., Makhmutov, V., Mathot, S., Nieminen, T., Onnela, A., Praplan, A. P., Riccobono, F., Rondo, L., Steiner, G., Tome, A., Walther, H., Baltensperger, U., Carslaw, K. S., Dommen, J., Hansel, A., Petäjä, T., Sipilä, M., Stratmann, F., Vrtala, A., Wagwww.atmos-chem-phys.net/17/14171/2017/ Atmos. Chem. Phys., 17, 14171–14180, 2017 14178 J. Lengyel et al.: Electron-induced chemistry in microhydrated sulfuric acid clusters ner, P. E., Worsnop, D. R., Curtius, J., and Kulmala, M.: Effect of ions on sulfuric acid-water binary particle formation: 2. Experimental data and comparison with QC-normalized classical nucleation theory, J. Geophys. Res.-Atmos., 121, 1752–1775, https://doi.org/10.1002/2015JD023539, 2016.

Hussein, T., Dal Maso, M., Petaja, T., Koponen, I. K., Paatero, P., Aalto, P. P., Hameri, K., and Kulmala, M.: Evaluation of an automatic algorithm for fitting the particle number size distributions, Boreal Environ Res, 10, 337-355, 2005.

---

## Author Comment (AC2) · 4 Jul 2019

We thank Referee #2 for his comments and suggestions, which we hope will help improving the manuscript. We have addressed the comments point by point below, with separate answers to the different topics/aspects of each of the main comments (1-3). In addition, the errorbars shown on Fig. 9.b were modified, as those did not correspond to the actual variability of $N_{50}$ absolute increase in the original version of the manuscript (former Fig. 8.b).

**Comment 1**: Section 2.2. The authors should discuss briefly the uncertainties and limitations of the equations 1 to 4 in calculating the particle formation (J) and growth (GR) rates in their data. **(A)** First, these equations have been developed originally for regional NPF, in which formation and growth of particles is assumed to take place relatively homogeneously over large spatial scales. This is apparently not the case in plumes where, among other things, various transport effects on J and GR should be taken into account. **(B)** Second, experimental limitations cause further uncertainties in determining J and GR. For example, using coagulation sink at 12 nm for all particles in the size range 12-19 nm in equation 1 causes some overestimation of coagulation losses, which results in underestimating J12. Also, Calculating J2 from J12 would require knowing GR in the size range 2-12 nm rather than that in the size range 12-19nm. While it is impossible to take into account the above issues to correct the data, the authors should at the very least discuss these issues briefly in section 2.2. If possible, the authors could also estimate whether resulting uncertainties are important or not with respect to their results.

**Reply 1**:

(A) This is an interesting discussion, as it is true that the abovementioned equations were originally derived to describe regional NPF. However, we do find a stronger increase of $N_{12-19}$ over time within the volcanic plume compared to non-plume conditions. The resulting increased particle formation rates observed on plume days indicate that particles are formed within the volcanic eruption plume "homogeneously" along the transport pathway to Maïdo. The same reasoning can be done with the growth rate. If nucleated particle appear to gradually "grow" with the typical banana shape, it means that the regional-type nucleation and growth process is taking place along the transport path. Based on these observations, we believe that Eq. (1-4) can be used to describe such events occurring at a sub-regional scale. This aspect is now better addressed in Section 3.1.3: *"Higher particle formation rates observed on plume days indicate that particles were constantly formed in the volcanic plume along the transport pathway to Maïdo, showing that nucleation and growth taking place over a distance of the order of 40 km appears like a regional scale homogeneous process, which can be described with the usual equations (Eq. 1-4) recalled in Section 2.2"*.

(B) In fact, the use of $CoagS_{12}$ for all particles in the investigated size range leads to some uncertainty in the calculation of $J_{12}$. However, based on Eq. (1), which is recalled below, we believe that $CoagS_{12}$ contributes to an overestimation (and not underestimation) of the particle formation rates.

$$J_{12} = \frac{dN_{12-19}}{dt} + CoagS_{12} \times N_{12-19} + \frac{GR_{12-19}}{7\,nm} \times N_{12-19} \qquad (1)$$

This is now clearly mentioned in Section 2.2: *« Note that the use of $CoagS_{12}$ for all particles in the range between 12 and 19 nm might cause some overestimation of the actual coagulation losses, and in turn lead to high estimates of $J_{12}$."* Also, we have included a sensitivity study at the end of Section 2.2 to give further insights into the effect of the particle growth rate in the determination of $J_2$ based on Eq. (2-4). All in all, this analysis reveals only limited effect of the growth rate variability/accuracy over the 2-19 nm in the conditions of our study.

**Comment 2:** Section 3.2.2. In this work, neither J2 nor H2SO4 concentration were measured directly, but were derived from other measured quantities, resulting in potentially large uncertainties in their values. This has implications which are not mentioned in the paper. **(A)** First, how reliable is the

observed relation between J2 and H2SO4 concentration, and how meaningful is it to compare this relation with those observed in studies were J and H2SO4 concentration were measured directly? **(B)** Second, how meaningful is it compare J obtained here with parameterized J due to binary water-sulfuric acid nucleation as a function of H2SO4 concentration? Does this comparison tell anything about nucleation mechanism?

**Reply 2**:

(A) We agree with the fact that the use of values derived from indirect calculations may lead to uncertainties, and such limitations should be kept in mind when interpreting the results. Regarding the determination of [$H_2SO_4$], we were unfortunately not able to evaluate the relevance of the proxy by Mikkonen et al. (2011) in volcanic plume conditions. Hence, in addition to the comment reported in Section 3.2.2 (*"which may stem from a reduced predictive ability of the proxy by Mikkonen et al. (2011) for the highest $SO_2$ mixing ratios"*), we have included another "warning" earlier in the manuscript, in Section 2.4: *"However, since the relevance of this proxy could not be evaluated in volcanic eruption plume conditions, neither from available measurements nor existing literature, one should keep in mind the potential limits of using such parametrization when interpreting the related results"*. Also, this demonstrates the need for additional measurements, as now clearly indicated at the end of Section 3.2.2: *"Such measurements would also allow more detailed evaluation of the proxy by Mikkonen et al. (2011) for the prediction of [$H_2SO_4$] in volcanic eruption plume conditions"*.

Concerning $J_2$, it goes without saying that the use of Eq. (2-4) naturally leads to some uncertainties in the calculation of $J_2$. However, these equations have already been approved and used in other studies, such as in the companion paper by Foucart et al. (2018). Moreover, the impact of $GR_{12-19}$ on the results derived from Eq. (2-4), which was rightly questionable, is now explicitly discussed in the revised version of the manuscript.

The results obtained in the present work were compared with that of other studies where measured values were used since those were the only "reference" we had for such comparison, in specific when restricting the literature to the poorly documented volcanic plume conditions. The comparison with Sahyoun et al. (2019) finally gave the chance to highlight the similarity of the values derived from the different approaches (measured *vs* calculated J and [$H_2SO_4$]) in comparable conditions, thus giving more credit to the values reported in the present analysis.

(B) Comparison with the theory has multiple interests, as mentioned in the paper in Section 3.2.2:
- Investigate whether or not the binary water-sulfuric acid nucleation can explain the particle formation rates reported in plume conditions;
- Show evidence for the ability of the parameterization to provide some reasonable estimates of the particle formation rates based on the knowledge of $SO_2$ mixing ratios only.

Based on the use of the parameterization, we have also included some additional discussions in the revised version of the manuscript regarding the contribution of the different nucleation mechanisms, charged and neutral (Section 3.2.2): *"As evidenced on Fig. 5.a, the total formation rate of 2 nm-clusters was mostly explained by ion induced nucleation for [H2SO4] below ~ $8×10^8$ cm-3, while neutral pathways seemed to explain the observations at larger sulfuric acid concentrations."*.

Finally, the fair agreement between the formation rates derived from measurements and that predicted by the theory gave additional support to our approach based on indirect measurements.

**Comment 3:** There are a few issues related to the particle growth that need some clarifications. **(A)** First, did the authors consider particle growth from one mode to another when estimating the relative contributions of primary and secondary particles in each mode? This remains a bit unclear when reading the results. **(B)** Second, the authors do not tell what were the typical air mass transport times from the volcano to the measurement site. This is important because for the reported particle growth rates (Fig. 2a), it takes a while before particles formed in the plume are able to growth into the Aitken mode, and for several hours before they can reach the minimum CCN size (assumed >50 nm here) or the accumulation mode. Is it feasible that particle formed by NPF in the volcanic plume reach these sizes

by the time measurements were conducted? **(C)** Third, while I agree with the authors that volcanic emissions are able to boost particle growth by e.g. heterogenous reactions of SO2 on particle surfaces, there seems to be some inconsistences in the storyline: on one hand the authors state that the plume appear not to influence the particle growth (section3.1.3), and on the other hand they state that particle growth in the plume increased both modal (section 3.3.1) and CCN (section 3.3.2) concentrations.

(A) The evaluation of primary vs secondary processes contributing to volcanic emissions were performed via the comparison of the 7:00 LT size distribution in and out of plume to evaluate primary emissions, and via the comparison of the maximum particle concentrations measured for each mode in plume and out of plume (observed between 11:00 and 14:00 LT depending on the modes and conditions) for the secondary particle contribution. The presence of a secondary contribution to the accumulation modes is likely the result of the growth of particles from the Aitken mode, due to the presence of more condensable gases. This is now more clearly stated in the text, and also recalled in the caption of Fig. 8.

(B) We have included a discussion regarding the growth of the particles nucleated close to the vent during their transport to Maido in Section 3.1.3, showing evidence for their ability to reach ~50 nm, i.e. CCN relevant sizes, over the distance of 39 km between the volcano and the station: *"A rough estimate for the transport time of the particles nucleated in the vicinity of the volcano to the Maïdo observatory can be obtained by dividing the distance between the sites by the median wind speed measured on NPF event days: 39 km ÷ 1.8 m s-1 ≈ 6 hours. This indicates that in such conditions, the GR_(12-19) reported on Fig. 3.a were often sufficient (> 8 nm h-1) for the newly formed particles (~ 1 nm) to grow up to CCN relevant sizes (~ 50 nm, see Sect. 3.3.2) during their transport, further explaining the observation of the typical banana shape of the events, both on plume and non-plume days. Similar analysis was repeated with the 75th percentile of the wind speed measured on NPF event days (2.9 m s-1), and, again, the observed growth was often fast enough (> 13 nm h-1) for the particles to reach 50 nm during the corresponding ~ 3 hours 45 minutes trip to Maïdo".*

However, when focussing more specifically on the CCN population in Section 3.3.2, we did not try to isolate the contribution of NPF to the observed CCN increase, as this would have been complex (impossible?), but we reported instead the contribution of secondary particles. Indeed, we did select the days when particles originating from NPF reached CCN-size, and in turn presumably contribute to CCN population, but the concentration increase that we calculated reflected the contribution of secondary processes as a whole, i.e. NPF but also growth of pre-existing particles, as mentioned in Section 3.3.2: *"On event days, the diurnal variation of $N_{50}$ was strengthened due to the concurrent formation of secondary aerosols, i.e. including the formation and growth of new particles as well as the growth of pre-existing larger particles mentioned earlier (Fig. 9.d)", "Following these hypotheses, the contribution of secondary aerosols to the observed CCN population was estimated from the difference between the median of the $N_{50}$ absolute increase observed on event days (i.e. resulting from transport of particles from the boundary layer and secondary aerosol formation) and that of non-event days (resulting from transport only)".*

(C) It is true that we report higher particle and CCN concentrations in plume conditions compared to non-plume days. However we do not relate those to increased particle growth rates, but rather to increased particle formation rates and to the presence of additional primary particles from volcanic origin during eruptive periods. This is for instance illustrated in the paragraph below, taken from Section 3.3.1: *"This observation was consistent with the enhanced production of particles previously reported for lower sizes in plume conditions, as the increase of the particles concentration in the Aitken mode most likely resulted from the growth of smaller particles originating from the nucleation mode. As already mentioned, the concentrations of the 2 accumulation modes measured at 07:00 LT were both significantly higher on plume days (235 4800 and 100 1300 cm-3, for the first and second accumulation mode, respectively) compared*

*to non-plume days (80 and 18 cm$^{-3}$, respectively), most likely due to additional sources of particles at the vent of the volcano during eruptive periods (see Section 3.2.1)"*.

Nonetheless it is true that the sentence at the end of Section 3.3.2 was confusing, and was thus slightly modified for clarity: "*the growth of  more particles to CCN relevant sizes was favoured during NPF events occurring in the presence of large amount of H2SO4 caused by the eruptions*".

**Minor/technical issues**

**Comment 1**: Page 7, line 4: "...when global radiation >50 ...". Something is missing from here (was?).

**Reply 1**: "was" added.

**Comment 2**: The format of providing the time difference (i.e. 2h10) in section 3.1.2 seems strange to me. Is this a correct way of expressing the time difference?

**Reply 2**: We are actually not sure about the expected format for expressing a time difference; we anyway changed it to "XX hours and YY minutes" instead of "XXhYY".

**Comment 3**: Page 10, line 2: "GR12-19 showed an important variability, ...". What do the authors mean by "important" here?

**Reply 3**: The "important" variability was related to the inter-quartile range; this aspect is now better addressed: "*GR$_{12-19}$ showed an important variability, as reflected by the monthly inter-quartile ranges, which were on average of the order of 80% of the corresponding medians*".

**Comment 4**: Excluding the last paragraph of section 4, the text in that section mainly summarizes the results discussed earlier in the paper. As a results, an appropriate title of this section would be "4. Summary and Conclusions".

**Reply 4**: Changed.

**Comment 5:** Would it be possible to change the lines and marks with yellow color in Figures into some other, more easily visible color?

**Reply 5**: The number of figures with yellow lines and/or markers is now limited in the revised version of the manuscript, since, after considering comments from Referee #1, strong-plume events are no longer highlighted in Section 3.3. For simplicity, and in order to optimize both the visibility and the contrast with other colours, we have decided to keep yellow in our colour code. However, we were able to improve Fig. 8 (yellow changed into purple), Fig. 7 (yellow changed into green) and Fig. 4.a (yellow numbers into black).

---

## Author Response (AR2)

We thank the Referee for checking the revised version of our manuscript and providing additional suggestions for further improvement. We have addressed the comments point by point below. Also, note that new figures, with higher resolution, have been prepared for the final version of the paper (and submitted in the .zip folder); because of this re-processing of the figures, the aspect of some of them might slightly differ from that of the older version, but we certify that their content / formatting is otherwise exactly the same as in the accepted version of the manuscript.

**Comment 1: A.** My only problem with the current manuscript is related to the detail of interpretation of the nucleation mechanism responsible for new particle formation (NPF) in the observed volcanic plumes. I agree with that the authors in that they have managed to demonstrate the crucial role of sulfur emitted by the volcano in causing NPF but, in lack of particle measurements below 10 (except ions) and no gaseous H2SO4 measurements, I really doubt anything concrete can be said about the actual nucleation mechanism involving H2SO4. **B.** Furthermore, the authors mention the important role of ion-induced nucleation at lower H2SO4 levels based on comparisons to a model parameterization. Do the authors have any support for this based on their observations? From AIS measurements, one could calculate the formation rate of 2 nm ions and this could be compared to the total formation rate of 2 nm particles obtained from equation 2. Admittedly, such a comparison would remain qualitative, rather than quantitative, but it might reveal the conditions under which ion-induced nucleation is expected to be most important (not just the level of H2SO4 concentration).

**Reply 1.A:**
We agree on the fact that robust conclusions cannot easily be drawn from the use of proxies, indirect measurements and model results only, but we do not pretend to provide a detailed interpretation of the nucleation mechanism that actually takes place: we do not interpret the coefficients of the relationship as a kinetic-type or activation-type nucleation mechanism. We observe a link between the calculated formation rate of clusters and the calculated $H_2SO_4$ concentrations, and we evaluate how our indirect observations compare with 1) the results from a recent study that uses direct measurements of sub-10 nm particles and $H_2SO_4$ (Sahyoun et al., 2019) and 2) those derived from an existing parameterisation of the formation rate (Määtänen et al., 2018). Our interpretation is that, given the uncertainty on our calculations, they agree with both measured and parameterized nucleation rates. We believe that, already in the initial version of the manuscript, the limits of our approach were clearly mentioned, but we further insist on the caution with which the reader should take our results in the revised version of the manuscript (See P11 L13-16, P18 L14-16 and L19-22 of the present document). Moreover, in order to make this message even stronger, we have added few comments and changed the phrasing of some sentences, regarding in specific the effect of the charge on cluster formation, as discussed below.

**Reply 1.B:** The calculation of 2 nm-ions formation rates would certainly help getting further insight into the role of ions in the nucleation mechanism in volcanic plumes. However, in absence of neutral cluster measurements below 3 nm, we were not able to provide such formation rates, as the knowledge of neutral clusters is needed to evaluate the attachment of ions on neutral clusters (Kulmala et al., 2012). While this term has been shown to contribute little to the formation of 2 nm ions in some environments (eg: Buenrostro Mazon et al., 2016), the lack of knowledge in volcanic plume conditions in particular prevents from the use of such hypothesis in the present work. This is now clearly stated in the revised version of the manuscript; also in absence of support from observational data, the phrasing of the conclusions regarding these aspects has also been changed, both in Section 3.2.2, in the abstract and in the conclusion (See P4 L10-14, P17 L31-34, P18 L1-6 and L30-32 and

P25 L20-22 of the present document).

**Comment 2:** The authors report an interesting, and rare, observation that the intensity of NPF is higher at larger values of condensation sink (CS) in volcanic plumes. I fully agree on the interpretations that the authors make on this. There is a very recent paper (Hakala et al. 2019, ACP 19, page 10555) where an exactly similar thing was observed in western Saudi-Arabia.

There, SO2 emissions from localized anthropogenic sources apparently caused NPF, but the overall idea seemed to be very similar to this paper: in plumes containing plenty of sulfur, NPF seems not be limited by the presence of pre-existing particles. The author could mention the study of Hakala et al (2019) shortly in their paper, especially since the results of these two papers seem to support each other strongly.

**Reply 2:** We thank the referee for pointing this reference, which is indeed of interest for our work. As suggested, it is now mentioned shortly in Section 3.2.1 (See P15 L12-15 of the present document).

**Minor comments:**

page 13, line 9: …injected into the atmosphere…: modified.

it is common to write: …, however,… (several places): modified.

it might be clearer to write "… gas-to-particle conversion…": modified.

**References:**

[revised manuscript text omitted]